# The impact of ancestral, genetic, and environmental influences on germline de novo mutation rates and spectra

O. Isaac Garcia-Salinas [1,6], Seongwon Hwang [2,3,6], Qin Qin Huang [1], Rashesh Sanghvi [1], Daniel S. Malawsky [1], Joanna Kaplanis [4], Matthew D. C. Neville [1], Felix R. Day [5], Raheleh Rahbari [1,7] ✉, Aylwyn Scally [2,7] ✉ & Hilary C. Martin [1,7] ✉

De novo germline mutation is an important factor in the evolution of allelic diversity and disease predisposition in a population. Here, we study the influence of genetically-inferred ancestry and environmental factors on de novo mutation rates and spectra. Using a genetically diverse sample of ~10 K whole-genome sequenced trios, one of the largest de novo mutation catalogues to date, we found that genetically-inferred ancestry is associated with modest but significant changes in both germline mutation rate and spectra across continental populations. These effects may be due to genetic or environmental factors correlated with ancestry. We find epidemiological evidence that cigarette smoking is significantly associated with increased de novo mutation rate, but it does not mediate the observed ancestry effects. Investigation of several other potential mutagenic factors using Mendelian randomisation showed no consistent effects, except for age at menopause, where factors increasing this corresponded to a reduction in de novo mutation rate. Overall, our study sheds light on factors influencing de novo mutation rates and spectra.

Germline mutation is a fundamental evolutionary process, and de novo germline mutations are a major cause of developmental disorders[1]. Such mutations occur at an exceptionally low rate[2,3], but this rate can be influenced by exposure to mutagens such as chemotherapeutic agents[4] and ionising radiation[5]. Tobacco smoke is known to affect the accumulation of de novo micro/minisatellites[6], but its effects on de novo point mutations, the best-characterised form of genetic variation, have not yet been studied. Genetic factors may also influence germline mutation rate[7]. For example, rare variants in DNA repair genes are known modifiers of somatic mutation rates and spectra[8–10], and have been shown to contribute to elevated rates of

germline mutation[4,10]. Certain common variants are associated with the rates of germline mutations at microsatellites[11], and common variants that decrease age at natural menopause, many of which implicate DNA repair genes, are associated with increased rates of de novo point mutations in the female germline[12]. Human polymorphism data indicate that mutation spectra have varied over time and between human populations[13,14], and a variety of genetic and environmental causes have been proposed for this[15]. However, the study of ancestral, genetic, and environmental factors influencing directly-measured germline mutations has been constrained due to the lack of appropriate datasets.

[1]Wellcome Sanger Institute, Wellcome Genome Campus, Hinxton, UK. [2]Department of Genetics, University of Cambridge, Cambridge, UK. [3]MRC Biostatistics Unit, School of Clinical Medicine, University of Cambridge, Cambridge, UK. [4]Genomics England, London, UK. [5]MRC Epidemiology Unit, Box 285 Institute of Metabolic Science, University of Cambridge School of Clinical Medicine, Cambridge, UK. [6]These authors contributed equally: O. Isaac Garcia-Salinas, Seongwon Hwang. [7]These authors jointly supervised this work: Raheleh Rahbari, Aylwyn Scally, Hilary C. Martin. ✉e-mail: rr11@sanger.ac.uk; aos21@cam.ac.uk; hcm@sanger.ac.uk

In this work, we explore the influence of continental-level genetic ancestry, parental genetics, and smoking behaviour on de novo point mutation rates and spectra by analysing up to 10,557 whole-genome-sequenced parent-offspring trios from the Genomics England 100,000 Genomes Project. Through the analysis of up to 688,948 de novo single nucleotide variants (henceforth referred to as "DNMs"), we show that genetic ancestry is significantly associated with differences in mutation rate and spectra. However, by estimating heritability in the best-represented ancestry in this cohort, we found that common genetic variants likely contribute little to DNM rate variation. We find that smoking is associated with a modest increase in de novo mutation rate and that this does not drive the ancestry associations. Our study provides insights into factors affecting DNM rates and spectra.

## Results

### Ancestral effects

We first explored whether the mutation rate and spectra differed between individuals of different genetic ancestries, using the continental-level genetic ancestry classifications produced by the 100,000 Genomes project[16,17]. These classifications are based on genetic similarity to individuals of known origins from the 1000 Genomes Project[18]. Although this classification does not fully capture the genetic diversity of human populations, it suffices for the main purpose of our analyses. For this analysis, we included 9820 trios for which both parents were inferred to come from the same continental-level ancestry group (198 African [AFR], 215 American [AMR], 53 East Asian [EAS], 8104 European [EUR], and 1250 South Asian [SAS]). First, we tested the association between ancestry and total DNM counts using generalised linear models, controlling for parental ages and technical covariates associated with the ability to call DNMs (Methods). We found that the AFR group is associated with a significantly higher number of DNMs compared to the AMR (fold change 1.042, adjusted $p = 1.62 \times 10^{-2}$), EUR (fold change 1.038, adjusted $p = 2.25 \times 10^{-3}$), and SAS (fold change 1.031, adjusted $p = 1.62 \times 10^{-2}$) groups (False Discovery Rate, FDR ≤ 5%, Supplementary Fig. 1; Supplementary Data 1). We estimate that the baseline DNM counts for trios in these groups (i.e. the expected trio DNM count if parental ages and all technical covariates have value 0) are 66.71

(±1.3, 95% CI), 63.99 (±1.25, 95% CI), 64.20 (±0.21, 95% CI), and 64.66 (±0.53, 95% CI), respectively (Fig. 1a).

As allele frequencies were not among the criteria to call or filter DNMs[4], these ancestry differences in DNM counts could not be driven by differences in the numbers of individuals from different ancestry groups in reference databases. However, in theory, these observed differences in DNM counts could be confounded by average differences in read mapping quality between ancestry groups due to reference bias[19] or differences in heterozygosity between groups[20], which might lead to different rates of missed heterozygous calls in parents. We controlled for several metrics and carried out various analyses to test whether these technical biases could be affecting our observed associations with ancestry (Supplementary Note 1; Supplementary Figs. 2–7). After removing DNMs with at least one read carrying the alternate allele in parents, the AFR/AMR and AFR/SAS differences became only nominally significant (Supplementary Fig. 5C). Overall, though, these sensitivity analyses suggested that our results were unlikely to be driven by technical artefacts. Similarly, a sensitivity analysis controlling for the number of protein-altering DNMs suggested that our findings were not the result of particular genetic ancestries being enriched for pathogenic DNMs, which could in theory happen if there were ancestry-correlated recruitment biases into the 100,000 Genomes Project (Supplementary Note 1, Supplementary Fig. 3).

We then tested whether ancestry was associated with differences in mutational spectra. For this, we divided the DNMs into seven categories according to which pyrimidine substitution was involved (and whether it was at a CpG site in the case of C > T transitions[21]), and then calculated the proportion of mutations in each category per trio (Methods). As pyrimidine substitution proportions are not independent from each other, we estimated ancestry effects using multinomial linear regression models and by comparing all possible ancestry pairs (Methods). When comparing the EUR and SAS groups, we identified significant differences (FDR ≤ 5%) in the proportion of C > A (EUR/SAS fold change = 0.93, adjusted $p = 3.11 \times 10^{-5}$), and CpG > TpG substitutions (EUR/SAS fold change = 0.94, adjusted $p = 9.21 \times 10^{-7}$). While using a more relaxed FDR threshold (FDR ≤ 10%), we also identified a significant differences in T > A proportions between the AMR and EAS

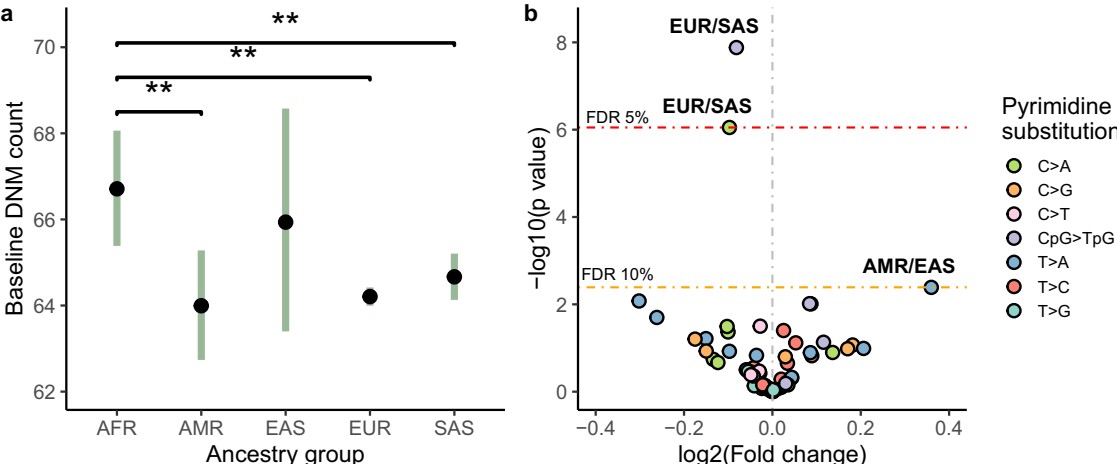

**Fig. 1 | Ancestry associations to DNM rate and DNM spectra. a** Estimated baseline DNM counts for offspring in each ancestry group. Estimates represent the intercept from a generalised linear regression model (Model 1, Methods), using the indicated ancestry group as the baseline in separate model runs. Bars indicate two-tailed 95% confidence intervals for the coefficient estimate. Asterisks denote significant DNM count differences between ancestry pairs after multiple testing correction (n tests = 10, FDR ≤ 5%) based on two-sided p-values. **b** Log₂ fold change estimates of pyrimidine substitution proportions for all pairwise ancestry group comparisons across seven substitution categories (n tests = 70). Estimates are derived from a compositional linear regression model (Model 2, Methods). Significant fold change differences, determined using two-sided p-values corrected for multiple testing (FDR), are labelled, with red and yellow lines indicating thresholds for FDR ≤ 5% and FDR ≤ 10%, respectively. Effect estimates correspond to the first ancestry in each label pair (e.g., C > A proportions in EUR are on average 0.98× smaller than in SAS). Analyses are performed on 9820 trios (n African [AFR] = 198, n American [AMR] = 215, n East Asian [EAS] = 53, n European [EUR] = 8104, n South Asian [SAS] = 1250).

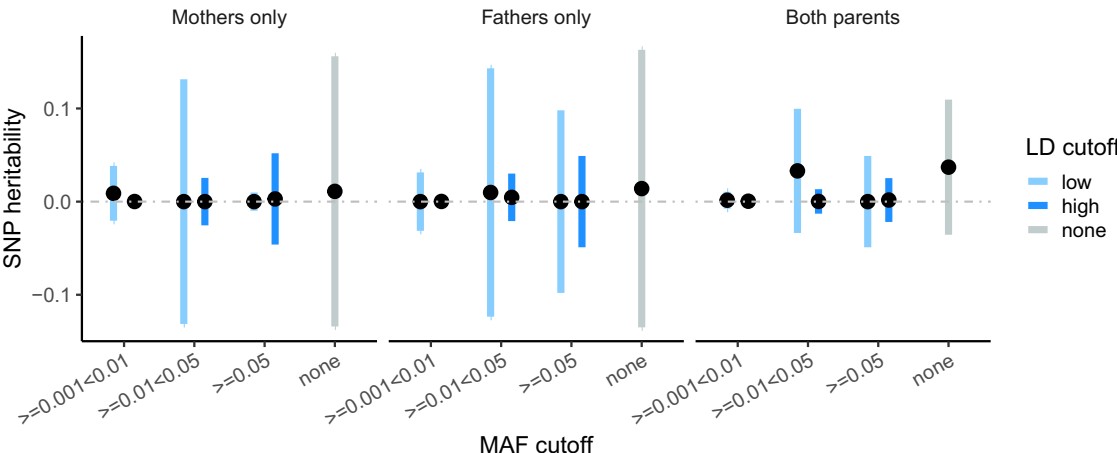

**Fig. 2 | SNP heritability estimates for DNM counts.** SNP heritability estimates (percentage of variance explained) for maternally phased DNMs in mothers ($n = 7993$, left), paternally phased DNMs ($n = 7892$, centre), and all phased DNMs across individuals ($n = 15,885$, right). Error bars represent two-tailed 95% confidence intervals for the heritability estimates. Estimates were calculated using the GREML-LDMS algorithm[25] (Methods) for variants stratified by minor allele frequency (MAF) and linkage disequilibrium (LD). Light and deep blue bars represent variants with LD scores below and above the genome-wide LD median, respectively. Grey bars indicate estimates for all variants in each subset, without MAF or LD filtering.

groups (AMR/EAS fold change = 1.28, adjusted $p = 9.54 \times 10^{-2}$) (Fig. 1b, Supplementary Fig. 8, Supplementary Data 2). The differences in C > A and CpG > TpG proportions observed between EUR and SAS groups recapitulate those seen using a different data set and a different method for quantifying population-specific germline mutation spectra based on polymorphism data[14] (Supplementary Note 2; Supplementary Data 3).

We also explored whether the distribution of DNMs across genomic loci varied by ancestry and mutational class (Supplementary Note 3, Supplementary Figs. 9–12). Regions of increased DNM incidence on chr8 and chr16, as seen in previous data[22] were driven primarily by maternal (particularly C > G) mutations, but no clearly significant differences between ancestries were evident.

Given that ageing is a major contributor to germline mutations[2,3], and that parental age at conception has varied across time and populations[23], it is plausible that differences in DNM counts and spectra may be influenced by ancestral variation in parental age. Although both parental ages at conception were included as a covariates in our models (Methods), we confirmed the reproducibility of our results by subsampling the cohort to remove parental age differences, finding that our associations remained consistent (Supplementary Note 1, Supplementary Fig. 7).

**Genetic effects**

Ancestry differences in DNM rates and spectra could reflect genetic differences between ancestry groups[24] that influence the germline mutation rate. Since detecting causal ancestry-stratified variants is likely to be difficult, we instead explored the contribution of common genetic variants to DNM rate within the largest ancestral group within our cohort, restricting to unrelated parents inferred to have European genetic ancestry (7786 mothers, 7692 fathers). We first estimated the variance in DNM rate explained by variants with minor allele frequency ≥0.1%, using GREML-LDMS[25]. For this analysis we used the parentally phased DNM counts previously produced by Kaplanis et al.[4]. After accounting for parental age, genetic variation captured by principal components, and technical factors (Methods), we did not obtain a significant SNP-heritability estimate with any of the minor allele frequency-linkage disequilibrium (MAF-LD) bin cutoffs tested, in either fathers, mothers, or both combined (Fig. 2). From this, we concluded that variance explained by common variants on DNM rate within this European-ancestry population must be too low to be detected in this sub-cohort. In spite of this finding, we note that potential effects of

ancestry-associated genetic variation on DNM rate may not be detected by this analysis.

We also ran GWAS on a broader sample including related European-ancestry individuals (7993 mothers, 7892 fathers) while controlling for relatedness using SAIGE[26]. This did not detect any genome-wide significant SNPs or indels ($p \leq 5 \times 10^{-8}$) in any sample subset (Supplementary Fig. 13).

**Environmental effects**

Ancestry differences in DNM rates and spectra could also reflect differences in environmental exposures between ancestry groups, which could include, for example, differences in diet or exposure to common mutagens such as cigarette smoke[27]. Very limited data on environmental exposures were available in the 100,000 Genomes Project, but we assessed the effect of smoking on DNM rate using the electronic health record (EHR) data (Methods). Using ICD10 codes, we created a binary "*ever smoked*" phenotype per individual. We then re-analysed associations between total DNM count and ancestry in a set of non-admixed trios in which at least one parent had ICD10 data available (192 AFR; 202 AMR; 46 EAS; 7389 EUR; 1204 SAS). For this, parental smoking was classified as: both parents smoke ($n = 293$), only the father smokes ($n = 664$), only the mother smokes ($n = 833$), or neither parent smokes ($n = 7243$). We obtained nearly identical ancestry effects to those reported in Fig. 1 (Supplementary Fig. 14), indicating that differences in parental smoking behaviour across ancestries are unlikely to be driving these ancestry associations. We observed significant effects of having a father who smokes or both parents smoking on total DNM count (Supplementary Fig. 15), with the caveat that these effect sizes may be noisy, since in many trios the smoking status of one parent was unknown and they were assumed to be a non-smoker. To refine our estimates of the smoking effect, we restricted to a set of 6599 fathers and 9133 mothers with the relevant subset of EHR data, and tested the effect of smoking on the number of DNMs derived from the relevant parent (i.e. phased DNM count, Methods). We found that having ever smoked was a significant predictor of increased DNM rate in females (fold change = 1.038, $p = 2.5 \times 10^{-2}$) and males (fold change = 1.019, $p = 4.3 \times 10^{-2}$), and in a sex-combined analysis (fold change = 1.024, $p = 3.6 \times 10^{-3}$, Fig. 3).

We attempted to identify differences in mutation spectra associated with parental smoking behaviour by applying different methods

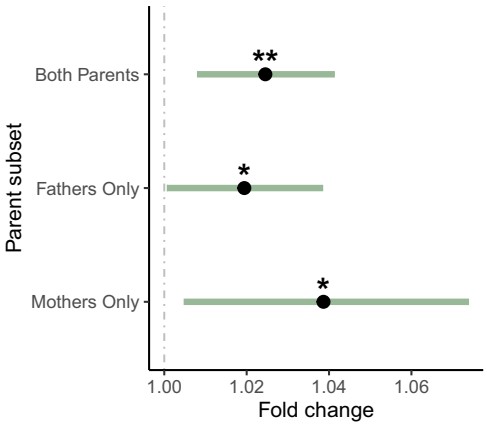

**Fig. 3 | Smoking effect on parental DNM rate.** Estimated effect (fold change difference) of ICD-10-derived smoking behaviour on phased DNM counts in all individuals combined ($n = 15{,}732$, $p = 3.6 \times 10^{-3}$), fathers ($n = 6599$, $p = 4.2 \times 10^{-2}$), and mothers ($n = 9133$, $p = 2.5 \times 10^{-2}$). Estimates correspond to the effect of smokers versus non-smokers in a generalized linear regression model (Model 6, Methods) within each subset. Smoking status was defined using ICD-10 codes Z58.7 ("exposure to tobacco smoke") and F17 ("mental and behavioural disorders due to tobacco use"), as described in the Methods section. Error bars indicate two-tailed 95% confidence intervals, and asterisks denote significance based on two-sided $p$-values: *$p \leq 0.05$, **$p \leq 0.005$.

and definitions of mutation spectra (Supplementary Note 4). We did not identify any significant associations between smoking and specific substitution types, and neither were we able to identify any known mutational signature[28] associated with smoking on DNMs (Supplementary Note 4, Supplementary Fig. 16).

We applied two-sample Mendelian Randomisation (MR) analyses to explore the influence of a range of factors on DNM rate, including some associated with reproductive traits. The exposures considered included age at natural menopause (ANM) (which we took as a positive control due to being already reported in the literature[12]), three smoking-related measures[29], alcohol use[29], body mass index (BMI)[29], and three traits chosen based on their associations with the top SNP from our DNM GWAS, namely sleep duration, hydrocele and spermatocele, and diseases of male genital organs (Methods). Following standard MR procedures[30], for each exposure of interest, we selected SNPs as instrumental variables based on a $p$-value threshold $5 \times 10^{-8}$ from publicly available GWAS, and applied LD clumping ($r^2 > 0.1$) to identify independent SNPs (Supplementary Data 4). Genetic factors contributing to late ANM were found to have a negative effect on DNM rate in mothers but not fathers, as previously reported in the same dataset[12] (Supplementary Fig. 17). Importantly, this was consistent across MR methods. We did not detect any further significant causal effect estimate for any of the remaining exposure traits, except that the number of drinks per week had a nominally significant causal effect on lower DNM count in fathers when using the simple median MR method (Supplementary Fig. 17). However, this result was not significant in any of the three other MR methods, and so should be treated with caution.

## Discussion

In this study, we investigated the influence of ancestral, genetic, and environmental factors on genome-wide germline de novo point mutation rates and spectra. Our findings suggest that ancestry correlates significantly with these molecular phenotypes, but the specific genetic and/or environmental factors driving these associations require further investigation.

First, we provided direct evidence that the DNM rate varies between individuals of different continental ancestries. Specifically,

the DNM rate is approximately 3–4% higher in individuals of African (AFR) ancestry compared to those of European (EUR), South Asian (SAS), and American (AMR) ancestries (Fig. 1a). A difference in DNM rate between EUR and SAS is also reflected in differences in mutational spectra between these groups (Fig. 1b). This mutational spectrum difference is characterised by a depletion of C > A and CpG > TpG mutations, similar to findings in a previous study[14]. However, unlike that study, we did not observe an enrichment in C > T and T > G mutations (Supplementary Note 2).

Interestingly, despite significant differences in overall DNM rates between AFR and non-AFR populations, no significant differences were found in their mutational spectra. This also contrasts with previous studies based on polymorphism data, where many of such differences were found[14]. This discrepancy may be due to the fact that polymorphism data reflect mutations accumulated over thousands of generations, whereas our data only capture mutations from a single generation. Historical germline mutation changes may be influenced by environmental or genetic factors that have since changed, resulting in contemporary populations showing no such spectral differences. Furthermore, our ability to detect differences in DNM rate and spectra across ancestry pairs was likely impacted by the sample size differences between groups in our cohort (Supplementary Note 5).

We note that the small average differences in DNM rate between ancestry groups that we have reported are likely to be exceeded by differences due to variation in the distribution of parental age at birth[31], which has changed across time[23] and has a much larger influence on DNM rate. Branch-length comparison between African and non-African populations has previously suggested an increase in mutation rate in non-Africans since the out-of-Africa event ~60 kyr ago[32]. Future studies may need to consider whether conclusions are changed by integrating these or other ancestry-associated differences in mutation rate and spectra into population genetic models[33], inferences about ancestry and demographic history (e.g. ref. [34]), and mutation rate models used for inferring genic constraint[35] and discovering genes enriched for DNMs in disease cohorts[1,36].

Second, we examined the contribution of genetic factors, particularly single nucleotide polymorphisms (SNPs), to variation in DNM rates. Although it is suspected that genetic factors play a role influencing the DNM rate[37,38], our analysis of SNP heritability within the European subset of the GEL cohort failed to detect any significant SNP heritability attributable to variants down to a frequency of 0.1%, consistent with a previous smaller study[39]. The observation that common SNPs associated with age at natural menopause are causally associated with DNM rate in mothers[12] (Supplementary Fig. 17) implies that there are common variants influencing DNM rate, but that they likely explain very little phenotypic variance. The error bars on our estimates in Fig. 2 imply that the heritability attributable to common SNPs with frequency greater than 5% is likely to be less than 15%. One possible explanation for the small contribution of common variants is the strong selection pressure against large-effect mutator alleles in the germline progenitors which prevents them from acquiring potentially deleterious new variants[40]. Future studies with larger sample sizes should investigate whether rare variants genome-wide explain an appreciable fraction of the variance in DNM rate. Further, larger sample sizes of diverse ancestral backgrounds will be needed to directly address whether ancestry-associated genetic variation significantly influences DNM rate.

Finally, we also explored environmental influences on DNM rate and spectra, with smoking emerging as a key factor. We have presented direct evidence that smoking is associated with the number of de novo point mutations in humans (Fig. 3), and that this effect is independent of that of ancestry (Supplementary Fig. 14). Specifically, we find that being a smoker increases the DNM count by approximately 2%, equivalent to less than one extra DNM per smoking parent over the reproductive lifespan (Methods). We emphasise that our

effect size estimates should be treated with caution since they rely on smoking behaviour being reported accurately in the health records, and it is plausible that these records are biassed towards recording heavy rather than occasional smoking behaviour. We also note that these associations do not show that smoking is causal for increased DNM rate, and indeed, both the MR and mutational signature analyses failed to provide evidence of this (Supplementary Fig. 17), although they may simply be underpowered; conceivably, smoking could be correlated with exposure to other mutagens. Further research is necessary to determine whether smoking or related exposures contribute directly to increased DNM rates.

This work demonstrates the imperative to include greater diversity of genetic ancestry in DNM studies, and to collect comprehensive epidemiological data on potential environmental mutagens, coupled with life-history information on participants. This will enable more powerful and robust investigation of the genetic and environmental influences on germline mutation. Although based on a larger cohort than previous studies, our analyses were nevertheless limited in their power to detect the effects of rare genetic variants, or more subtle differences across ancestries and sexes on DNM rate variation. Since existing methods for detecting and estimating genetic associations rely on genetically homogeneous cohorts, future studies may also require methodological developments to effectively disentangle ancestry-correlated genetic variants from environmental factors affecting de novo mutation rates and patterns.

## Methods

### Description of whole-genome de novo mutations and phasing information across a cohort of 13,949 family-trios

We used readily available data from the rare disease cohort of the 100,000 Genomes Project (100kGP) generated by Genomics England[16,41]. The 100kGP was approved by the East of England−Cambridge Central Research Ethics Committee (REF 20/EE/0035). Studied families were selected on the basis of having at least one offspring with an undiagnosed rare disease[16,41]. Although 13,949 trios were originally recruited as part of this project, the data freeze we accessed (v16) contained information for just 12,017 trios due to some individuals having withdrawn their consent to participate. De novo variant calls (DNMs) per trio were generated using the Platypus variant caller[42] based on multi-sample VCFs that were generated at the trio level (https://re-docs.genomicsengland.co.uk/de_novo_data/). DNMs included single nucleotide variants (SNVs, $n = 906{,}643$) and short insertion-deletion events of up to 250 bp (indels, $n = 72{,}481$), which were previously filtered using a stringent criteria that is fully described elsewhere[4]. This dataset included only mutations from autosomes (chr1-22) and chrX, but did not include DNMs from chrY and mitochondria. De novo SNVs were phased to the parent of origin using parental heterozygous single nucleotide polymorphisms found in a 500 bp vicinity of the reported DNM. This strategy allowed phasing of ~26% of the original DNMs, out of which 80% phased to the paternal germline[4]. The parental phasing process excluded DNMs on chrX. Downstream analyses were focused on de novo SNVs, and we refer to these as DNMs for simplicity.

### Genetically inferred ancestry information

Ancestry information for all of the participants was readily available in the Genomics England (GEL) research environment. This resource was produced from joint principal component analysis (PCA) from the 100kGP individuals and the 1000 Genomes Project phase 3 (1kGP3)[16,17]. Briefly, PCs for the 1kGP3 were calculated using a set of 188,382 high quality SNPs (MAF ≥ 0.05) intersecting with the 100kGP, in unrelated individuals. The 100kGP data was then projected onto 1kGP3 PC loadings, and a random forest classifier based on the first 8 PCs and continental-level ancestry classifications from 1000 Genomes was used to predict ancestry for each individual in the 100kGP dataset.

Ancestry classifications from 1000 Genomes corresponded to one of the five continental-level super-populations: African (AFR), American (AMR), European (EUR), East Asian (EAS), and South Asian (SAS)[16,18]. Each individual in the 100kGP cohort was assigned a probability of belonging to each of these populations.

Finally, unrelated individuals assigned to each population with a probability ≥0.8 were used to calculate population-specific PCs[17]. European-specific PCs were used in the heritability estimation and GWAS, which were focused on individuals inferred to have European ancestry.

### Sample filtering

We removed nine trios in which the proband was previously identified as having a significantly elevated rate of DNMs (hypermutator individuals)[4], to prevent spurious associations in our study due to their elevated DNM rate and characteristic DNM spectra. Some families in this study included multiple offspring. As the DNM rate increases linearly with age[2,3], we kept the trio corresponding to the youngest sibling in these multiplex families to increase our chance of observing DNM events. We further removed all trios with ≥1 individuals missing metadata on date of birth, sequencing statistics, or de novo mutation calling Bayes factors. This left 10,557 trios and 688,948 DNMs for analysis.

### Associations between ancestry and DNM rate

For this analysis, DNM rate was defined as the total DNMs detected per trio (i.e. unphased DNMs). For ancestry, we relied on the classifications readily available in the GEL research environment (described above). For each individual, we took the ancestry assignment with the highest probability (see previous section). We further filtered out trios with parents assigned to different ancestry groups, leaving 9820 non-admixed trios and 639,361 DNMs for analysis.

DNM counts are associated with parental sex and age at conception[2]. Apart from these biological factors, technical variables associated with sequencing quality metrics in the family trio have been shown to explain up to 1.3% of the variance of DNM counts in this cohort[4]. These metrics include sequencing coverage metrics such as: the mean sequencing depth (per trio member), the percent of aligned reads (per trio member), as well as trio-level summary statistics such as the median Bayes factor (from the Platypus variant caller) and the median variant allele fraction (i.e. alternative allele to total site sequencing coverage ratio, [VAF]) of all DNMs called in a given trio[4]. All of these biological and technical covariates are accounted for in our models.

Associations were run using generalised linear models (R glm() function) from the quasi Poisson family (logit link), (selected to account for the mean-variance overdispersion of the phased DNM count data, Supplementary Note 7), as shown in Model 1:

$$
\begin{aligned}
trio\ DNM\ counts = {} & \beta_0 + trio\ ancestry \cdot \beta_1 + maternal\ age\ at\ conception \cdot \beta_2 \\
& + paternal\ age\ at\ conception \cdot \beta_3 + mean\ sequencing\ depth_{mother} \cdot \beta_4 \\
& + mean\ sequencing\ depth_{father} \cdot \beta_5 + mean\ sequencing\ depth_{offspring} \cdot \beta_6 \\
& + percent\ aligned\ reads_{mother} \cdot \beta_7 + percent\ aligned\ reads_{father} \cdot \beta_8 \\
& + percent\ aligned\ reads_{offspring} \cdot \beta_9 + median\ Bayes\ Factor\ per\ trio \cdot \beta_{10} \\
& + median\ VAF\ per\ trio \cdot \beta_{11} + \varepsilon
\end{aligned}
$$

Here, *trio ancestry* is a factor covariate taking one of 5 possible identities (AFR, AMR, EAS, EUR, SAS), and *median VAF per trio* corresponds to the median offspring variant allele fraction of all high quality DNMs found in such an offspring. Prior to running associations, we scaled and centred all numeric covariates to have a mean zero using the scale() function from base R. As ancestry is a five-level factor covariate, a single ancestry was used as a baseline to compare against all of the remaining four. We iteratively changed this baseline until we obtained all possible, non-redundant, pairwise ancestry comparisons ($n = 10$). We took the exponential of each ancestry coefficient to obtain

fold change effect size estimates (FCs). In this context, FCs represent the change associated with being from ancestry $A$ versus ancestry $B$. All $p$-values associated with ancestry coefficients were corrected to account for multi-testing using the R p.adjust() function, and the "fdr" method. All results with an adjusted $p \leq 0.05$ were deemed to be significant. Baseline DNM count estimates per ancestry were obtained by exponentiating the intercept term in the model where the ancestry of interest was used as the baseline.

## Associations between ancestry and DNM spectra

The mutational spectrum was defined as follows. Each DNM was classified according to the pyrimidine base of the Watson-Crick base pair, which allows for a standardised way to identify and compare mutational patterns[43]. In this way, each mutation can be classified as one of six possible pyrimidine substitutions (i.e. C > A, C > G, C > T, T > A, T > C, T > G). In addition to these, we differentiated between C > T substitutions occurring on CpG sites (i.e. CpG > TpG), and those occurring elsewhere (i.e. C > T). This was done to account for the increased number of transitions occurring at CpGs due to spontaneous cytosine deamination[2]. CpGs were defined as Cytosine bases followed by a Guanine base in the 5′ to 3′ direction[44]. Hence, we annotated each DNM as one of 7 possible pyrimidine substitutions, counted the occurrence of each substitution per trio, and obtained the proportion of each substitution out of the total DNMs per trio.

In a similar way to DNM counts, we modelled pyrimidine proportions as a function of trio ancestry, as well as biological and technical covariates associated with DNM mutation calling (Model 2).

$$pyr_y = \beta_0 + trio\ ancestry \cdot \beta_1 + maternal\ age\ at\ conception \cdot \beta_2$$
$$+ paternal\ age\ at\ conception \cdot \beta_3 + mean\ sequencing\ depth_{mother} \cdot \beta_4$$
$$+ mean\ sequencing\ depth_{father} \cdot \beta_5 + mean\ sequencing\ depth_{offspring} \cdot \beta_6$$
$$+ percent\ aligned\ reads_{mother} \cdot \beta_7 + percent\ aligned\ reads_{father} \cdot \beta_8$$
$$+ percent\ aligned\ reads_{offspring} \cdot \beta_9 + median\ Bayes\ Factor\ per\ trio \cdot \beta_{10}$$
$$+ median\ VAF\ per\ trio \cdot \beta_{11} + \varepsilon$$

where $pyr_y$ represents the per trio proportion of DNMs being classified in the $y$ category (i.e. either C > A, C > G, C > T, T > A, T > C, T > G, or CpG > TpG). To account for the interdependence of pyrimidine proportions, we used the compositional data modelling approach from Zhou et al.[45], implemented in the "MicrobiomeStats" R package via the linda() function. This method models centred log-ratio transformed proportions using linear regression and applies bias correction to effect estimates and $p$-values to account for the compositional nature of data, where components are interdependent and constrained to sum to a constant[45].

As before, we iteratively changed the trio ancestry baseline until we had tested all possible ancestry combinations ($n = 10$). $P$-values associated with each ancestry coefficient were corrected to account for multi-testing (using the p.adjust() function and the "fdr" method) by pulling together all pyrimidine substitution tests ($n = 70$ tests).

## Derivation of residualised phased DNM counts for genetic associations

We obtained phased autosomal DNM counts ($n = 152,376$) across a cohort of 15,885 genetically identified European parents ($n$ fathers = 7892; $n$ mothers = 7993; EUR probability ≥0.8 previously produced by Kaplanis et al.[4]. In addition to the same technical and biological factors accounted for in previous models, we further included the number of SNVs per trio as this would affect the ability of phase DNMs due to the method used for this process[4]. Hence, as our phenotype, we took the residuals from a generalised linear model from the quasi-Poisson

family (Model 3 - sex combined):

$$phased\ DNM\ counts = \beta_0 + sex \cdot \beta_1 + parental\ age\ at\ conception \cdot \beta_2$$
$$+ sex{*}parental\ age\ at\ conception \cdot \beta_3 + mean\ sequencing\ depth_{mother} \cdot \beta_4$$
$$+ mean\ sequencing\ depth_{father} \cdot \beta_5 + mean\ sequencing\ depth_{offspring} \cdot \beta_6$$
$$+ percent\ aligned\ reads_{mother} \cdot \beta_7 + percent\ aligned\ reads_{father} \cdot \beta_8$$
$$+ percent\ aligned\ reads_{offspring} \cdot \beta_9 + total\ SNVs_{mother} \cdot \beta_{10} + total\ SNVs_{father} \cdot \beta_{11}$$
$$+ total\ SNVs_{offspring} \cdot \beta_{12} + median\ Bayes\ Factor\ per\ trio \cdot \beta_{13}$$
$$+ median\ VAF\ per\ trio \cdot \beta_{14} + \varepsilon$$

To attempt to identify sex specific differences on DNM rate, we residualized phased DNM counts in a similar way to Model 3 while subsetting the dataset to either fathers or mothers alone (Model 4 - sex specific fathers; Model 5 - sex specific mothers), dropping the sex term and the sex * age at conception interaction term.

Residuals obtained from models 3–5 were later used for heritability estimation and genome-wide association analysis.

## SNP heritability estimation

Genotypes for heritability estimation were processed as follows. Aggregated genotypes from 100kGP project participants had been previously masked to set as missing sites with sequencing depth (DP) < 10, genotype quality (GQ) < 20, or heterozygous genotypes failing the binomial test for allele balance with a $p < 10^{-3}$[46]. We restricted the heritability analysis to the 15,885 EUR parents mentioned in the previous section, from which we further removed 478 related individuals (up to 3rd degree) that were previously identified in GEL via the KING genetic relatedness algorithm[16,47]. From these files, we further removed genotypes with a missing rate >0.02 and with a Hardy–Weinberg equilibrium test $p < 10^{-6}$.

Heritability was estimated using the residualized phased DNM counts from models 3 to 5 described in the phased DNM residualisation section (i.e. both parents combined, fathers alone, and mothers alone). Each individual phenotype subset was matched to its respective genotype subset (e.g. fathers-only residualized DNMs to fathers-only genotypes) and 20 population-specific PCs (EUR) were included in the heritability estimation regression for all methods[16].

GREML-LDMS was used to calculate heritability across six variance components[25]. For this, genetic relatedness matrices (GRMs) were calculated using three MAF bins (MAF ≥ 0.001 & <0.01, MAF ≥ 0.01 & <0.05, and MAF ≥ 0.05), and two LD bins (low and high LD). LD bins were derived from LD scores calculated directly from the genotype data mentioned above for all variants with MAF ≥ 0.01 in a 200Kb window. Low LD was defined as those variants having a LD score lower than the genome wide median (median LD score = 81.31), or higher than this in the case of high LD.

## Genome-wide association study for DNM rate

We ran a genome-wide association study for DNM rate on common autosomal SNVs and indels (MAF ≥ 0.05). For this we used the residualized phased DNM counts (models 3–5 in the DNM residualization section) and the linear mixed model implementation of SAIGE v1.0.7[26]. Genotype data corresponded to that in the heritability estimation section (including related individuals), with three individual subsets: mothers only ($n = 7993$), fathers only ($n = 7892$), and both parents combined ($n = 15,885$). As before, each phenotype subset (residualised phased DNMs) was matched to its corresponding genotype subset.

## Analysis of smoking and DNM rate

We derived a proxy for the binary "ever_smoked" phenotype (0|1) using ICD10 codes available in the secondary care (admitted patient care - APC) data as part of the hospital episode statistics (HES) records available for 100kGP participants[16]. We identified individuals with at least one ICD10 code related to tobacco smoking behaviour. Specifically, we included Z58.7 ("Exposure to tobacco smoke") and F17-

derived codes ("Mental and behavioural disorders due to use of tobacco"), with most of the records falling under F17 codes ($n$ F17 codes = 2165; $n$ Z58 codes = 4). We note that F17 was renamed as "Nicotine dependence" in the 2024 version of ICD10, and that the code Z72, representing "tobacco use", was not present in the accessed electronic health records. We classed individuals having ≥1 smoking ICD10 entry as smokers (i.e. ever_smoked = 1) or non-smokers (i.e. ever_smoked = 0).

With this, we first built an integrative model of DNM rate including both smoking and ancestry. From the set of non-admixed trios ($n$ trios = 9820), we further kept trios where at least one parent had APC data available ($n$ trios = 9033). From these, in 6223 trios both parents had APC data available, 205 had it for the father only, while 2605 had it for the mother only. Together with the individual level "ever smoked" annotation produced before, for each trio we encoded a 4 level factor called "parental smoking". This indicates whether both parents smoked ($n$ = 293), only the father smoked ($n$ = 664), only the mother smoked ($n$ = 833), or none smoked ($n$ = 7243). Given that APC data was available for a single parent in 2810 trios, our "parental smoking" encoding assumes that the parent missing APC information is a non-smoker. We re-ran Model 1 while adding the "parental smoking" covariate.

### Refining smoking effect estimate using phased DNM data

From 20,245 non-hypermutator parents with phased DNM information (previously produced by Kaplanis et al.[4]), and complete metadata we were able to retrieve at least one ICD10 entry for 15,732 individuals ($n$ fathers = 6599 fathers; $n$ mothers = 9133). Out of 15,732 individuals with ICD10 code information, 2169 had at least one ICD10 entry relating to nicotine dependence from ICD10 codes Z58.7 or F17 ($n$ fathers = 996 fathers; $n$ mothers = 1173).

Combining data from mothers and fathers, we ran associations using a generalised linear model of the quasi-Poisson family (R glm() function), where we controlled for covariates that affect both the calling and phasing of DNMs (Model 6).

$$phased\ DNM\ counts = \beta_0 + ever\ smoked_{(0|1)} \cdot \beta_1 + sex \cdot \beta_2$$
$$+ parental\ age\ at\ conception \cdot \beta_3 + parental\ sex*parental\ age\ at\ conception \cdot \beta_4$$
$$+ mean\ sequencing\ depth_{mother} \cdot \beta_5 + mean\ sequencing\ depth_{father} \cdot \beta_6$$
$$+ mean\ sequencing\ depth_{offspring} \cdot \beta_7 + percent\ aligned\ reads_{mother} \cdot \beta_8$$
$$+ percent\ aligned\ reads_{father} \cdot \beta_9 + percent\ aligned\ reads_{offspring} \cdot \beta_{10}$$
$$+ total\ SNVs_{mother} \cdot \beta_{11} + total\ SNVs_{father} \cdot \beta_{12} + total\ SNVs_{offspring} \cdot \beta_{13}$$
$$+ median\ Bayes\ Factor\ per\ trio \cdot \beta_{14} + median\ VAF\ per\ trio \cdot \beta_{15} + \varepsilon$$

We note that this and other models exclude any effects of cross-parental age on phased mutations (e.g. an effect of maternal age on paternally phased DNMs). This exclusion is justified by the analysis of cross-parental effects presented in Supplementary Note 6.

Finally, we ran associations separately for each parent. In such cases we dropped the sex and sex*age covariates while subsetting to either mother or fathers each time. We quantified the effect of smoking in fathers relative to a given change in paternal age using using the formula 4.385 + 1.296*paternal age, where the intercept and slope were obtained from a simplified negative binomial model keeping all of the covariates in Model 6 but dropping the smoking effect.

### Mendelian randomization

To determine causal relationships between different exposures and DNM rate, we ran two-sample Mendelian randomization (MR) analysis using the inverse-variance weighted, MR Egger, and the simple and weighted median-based approaches[30]. Instrumental variables for the putative exposures were selected from publicly available GWASs. The selected risk factors and their respective sources are shown in Supplementary Data 4. These were:

- Age at natural menopause, selected on the basis of the results in Stankovic et al.[12], and considered a positive control.
- Smoking initiation, smoking cessation and age of smoking initiation, chosen based on epidemiological associations between smoking and male infertility[29] and based on our own results which suggest an association between smoking and DNM rate.
- Alcohol use (drinks per week), chosen based on epidemiological associations between alcohol use and male infertility[29].
- BMI, chosen based on its association with infertility[48], which is thought be also associated with DNM rate[49].
- Three phenotypes chosen through a phenome-wide association study of the top SNP from our sex-combined DNM rate GWAS (rs71599241, $p$-value = $1.01 \times 10^{-7}$) conducted using an atlas of genetic associations in UK Biobank[50]. We selected three phenotypes that were nominally significantly associated with this SNP and seemed relevant to reproduction: hydrocele and spermatocele (ICD10 code N43) ($p$ = 0.001), diseases of male genital organs (ICD10 codes N40-51) ($p$ = 0.004), and sleep duration ($p$ = 0.009)[29].

We note that phenotypes such as age at natural menopause and BMI represent latent variables encompassing biological processes likely affecting the germline mutation, rather than affecting the germline mutation themselves. For example, age at menopause is thought to reflect the efficiency of the DNA repair machinery, which itself likely influences germline de novo mutation[12]. In this sense, the strict MR assumption of no pleiotropic influence of the instruments on the outcome does not hold, and the measured effect reflects genetic correlation rather than causality.

As linkage disequilibrium (LD) between instrumental variables can bias MR causal effect estimates due to horizontal pleiotropy[30], we pruned the instrumental variables (LD $r^2 > 0.1$). The total number of instrumental variables used per risk factor and LD threshold is shown in Supplementary Data 4. The effects of these variants on paternal, maternal, or sex-combined DNM rate were estimated in the GWASs described in the previous section. The full summary statistics for all exposure phenotypes and parameter combinations used for this analysis are included in Supplementary Data 5 for paternal DNM rate, and Supplementary Data 6, for maternal DNM rate, and Supplementary Data 7 for the sex-combined DNM rate.

### Reporting summary

Further information on research design is available in the Nature Portfolio Reporting Summary linked to this article.

## Data availability

Whole-genome sequence data and phenotypic data from the 100,000 Genomes project can be accessed by application to Genomics England (https://www.genomicsengland.co.uk/join-us). Key files of DNMs and covariates used for the regression analyses in this paper are available within the Genomics England research environment (/re_gecip/shared_allGeCIPs/aeg_dnm_2024). GWAS summary statistics of DNM rate generated in this study (Supplementary Data 11–13) have been deposited in figshare (https://doi.org/10.6084/m9.figshare.28633352) and the GWAS catalogue (https://www.ebi.ac.uk/gwas/home). GWAS catalogue accession numbers are: GCST90565198 (sex combined DNM rate), GCST90565197 (paternal DNM rate), and GCST90565196 (maternal DNM rate). Publicly available GWAS summary statistics can be accessed at various resources: http://geneatlas.roslin.ed.ac.uk, https://conservancy.umn.edu/handle/11299/241912, https://www.reprogen.org/. Somatic mutations from ascertained smoker individuals can be accessed at: https://data.mendeley.com/datasets/b53h2kwpyy/2. Reference single base substitution mutational signatures used for deconvolution can accessed at: https://cancer.sanger.ac.uk/signatures/sbs/.

## Code availability

We have deposited a copy of the code used to generate our regression analyses in github (https://github.com/isaacg322/EnvGenDNM). This code is also available and fully reproducible within the Genomics England research environment (/re_gecip/shared_allGeCIPs/aeg_dnm_2024).

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

## Acknowledgements

This research was made possible through access to data in the National Genomic Research Library, which is managed by Genomics England Limited (a wholly owned company of the Department of Health and Social Care). The National Genomic Research Library holds data provided by patients and collected by the NHS as part of their care and data collected as part of their participation in research. The National Genomic Research Library is funded by the National Institute for Health Research and NHS England. The Wellcome Trust, Cancer Research UK and the Medical Research Council have also funded research infrastructure. We also thank Prashant Gupta and Federico Abascal for their valuable feedback and time for discussion. This research was funded in part by Wellcome (grant no. 220540/Z/20/A, "Wellcome Sanger Institute Quinquennial Review 2021–2026"). For the purpose of open access, the authors have applied a CC-BY public copyright licence to any author accepted manuscript version arising from this submission.

## Author contributions

O.I.G.S. and S.H. conducted the analyses and drafted the manuscript. Q.Q.H., D.S.M., J.K., M.D.C.N., R.S. and F.R.D. advised on QC or specific analyses. R.R., A.S. and H.C.M. supervised the study.

## Competing interests

J.K. is an employee of Genomics England. The remaining authors have declared no competing interests.
