## [Transparent Peer Review file · Nature Communications]

The impact of ancestral, genetic, and environmental influences on germline de novo mutation rates and spectra

Corresponding Author: Dr Aylwyn Scally

Version 0:

Reviewer comments:

Reviewer #1

(Remarks to the Author)

DNA mutation rate is a quantitative trait that can vary between cells and individuals, and, and it is understood that this trait is influenced by genetic variation in multiple species. In this study by Garcia-Salinas, et al, use one of the largest datasets of germline mutation assembled to investigate environmental and genetic influences on mutation rate in humans. The authors find associations of ancestry, smoking, and age of menopause on de novo mutation rate. Overall I find the manuscript to be well written and the analyses to be rigorous and thorough. I only have minor comments.

Minor comments:

Could the authors please describe in the methods whether the sex chromosomes and mitochondrial genome were included in this analysis? That information may be contained in Kaplanis et al 2022 but it would be helpful for the reader not to have to revisit that.

I found references to model 1,4,5,6,7, and 8. But there doesn't seem to be a model 2 or model 3?

The author use a number of technical and biological covariates in their mutation models (i.e. Models and 4-8 referenced above). Can they briefly explain why these covariates were selected, and whether there is any statistical evidence that each of these covariates influences DNM count? In other words, I'm looking for justification of the inclusion of the "nuisance" parameters so to speak, not the main effect (hypothesis-driven) parameters (like "ever smoked"). If this has already been explored in Kaplanis et al 2022 that is fine, but it would be good to explicitly state this in the methods (i.e. that each of these terms has been demonstrated to influence total observed DNM).

Line 101 "The differences in C>A and T>C proportions recapitulate what has been previously reported [14]). I don't think this sentence fully captures what is going on here and led me to spend some time on Harris and Pritchard until I read the Supplementary Note 3. I would recommend something like: "The differences in C>A and T>C proportions seen between EUR and SAS are recapitulated using a different data set and a different method for quantifying population-specific germline mutation spectra [14])."

Line 246 the authors sound a call to collect "comprehensive epidemiological data on potential environmental mutagens". To that end, can the authors describe what type of information is available on the geographic sampling location of the subjects in this study? Is there any sort of GPS encoding of the subject's present day or historic domicile(s)?

Line 539 the URL is incorrect; (<https://www.genomicsengland.cgfbo.uk/research/academic/join-gecip>) should be "co.uk" not cgfbo.uk

Supplementary Figure 8: The legend could be clarified a bit. I believe this figure is a cartoon showing properties of three different distributions plotted against simulated data (grey circles). What type of data are the grey circles meant to represent? Why are some circles larger than others? I assume the symbolism is meant to be analogous to the binned mutation data shown in Supp Fig 9A and 9B. But I don't want to assume too much.

(Remarks to the Author)

Remarks to the Author:

De novo mutation (DNM) rates are an important 'meta' phenotype because of their influence on many other biological processes, including evolution, disease and ageing. However, as the authors of this manuscript note, it's unusually difficult to study DNMs and their associated properties, because detecting them requires genomic information from parents of the samples in addition to the samples themselves (i.e. trio data).

The authors of this manuscript take advantage of a large dataset of trios and DNM calls previously published by Kaplanis et al, 2022 to conduct a variety of biostatistical analyses about DNM rates. The direct associations between tobacco exposure and DNM rates presented in this manuscript are entirely novel, as are some of the results about DNM rate and spectrum differences between people of different continental ancestries. Because the dataset is considerably larger than those in comparable studies, the authors are also able to readdress several longstanding scientific questions that previous researchers had insufficient power to answer. For instance, a prior SNP heritability analysis from Kessler et al. 2020, cited by these authors, uses around ~90 K DNMs (nearly 10 times fewer than the present work) from around 1500 trios (~5 times fewer than the present work).

The methodology is largely sound and thoroughly described, though some figures have ambiguous interpretations and should be redesigned (see "minor comments" for details). The conclusions are generally justified and appropriately cautious. Given its combination of novel results and timely reanalyses, we think this manuscript usefully pushes forward the state of knowledge on causes of DNM rate and spectrum variation within the general population.

Specific comments:

1. The content needs to be restructured to make the overall scientific narrative clearer. At present, the manuscript presents many different analyses in sequence without much commentary on how they relate to each other. The clarity and likely impact could be increased by re-organising the "Results" to reflect the three different factors in the title (ancestral, environmental and genetic influences on DNM rates). A section on "ancestral influences" could cover all the results about DNM rates and spectra between different continental groups. These results motivate the rest of the manuscript: the observation that there *are* systematic differences in DNM rates and spectra across both families and ancestries indicates there may be genetic and environmental influences on DNM rates that we are yet to find. A section on "genetic influences" could cover the SNP heritability results and GWAS results, and a section on "environmental influences" could cover the smoking and Mendelian randomisation analyses.

2. After re-organising the "Discussion" section to match this structure, it would also be useful to explicitly mention the ways in which the present study was not able to directly address some of the questions posed in the introduction. For instance, the low SNP heritability in EUR samples doesn't necessarily mean there aren't genetic influences on DNM rates that act between ancestral groups (or within the many ancestral groups that weren't tested in the present study). In addition, the authors should clearly point out that uneven sample sizes between continental groups likely has a large impact on which pairs of populations are found to have significantly different mutation rates and spectra.

3. Figures 1A and 1B are very difficult to parse: the reader has to look both across a row and up a column to see all the cells relevant to a certain continental population, and they then have to think about how the model reference changes across these cells to interpret the numbers inside them. It should be made more obvious which group is elevated with respect to which other group—for example, does an "excess" mean the "row" group is elevated over the "column" group or vice versa? We suggest re-running these analyses using a common model reference in each case ('EUR', perhaps), and then presenting the results with a barchart instead of a heatmap (similar to Supplementary Figure 2). This presentation would also allow confidence intervals to be displayed, and so would help the reader get a sense of which results might fall below statistical significance simply because of sample size (those involving the EAS population, for instance).

4. The finding of significant mutation spectrum differences between human continental groups is one of the most interesting findings of the paper given that this has been long suspected based on segregating variation but was not detectable using previous smaller DNM datasets. Given this, the paper could benefit from additional efforts to interpret these differences. Do the authors think that the finding of no significant differences between AFR and other groups is mainly driven by the small sample size of the African trios, or is the sample size sufficient to rule out the magnitude of effects predicted by the Harris and Pritchard study? Are the observed differences among the mutation spectra of continental groups explainable by differences in the distribution of parental ages, as recently suggested by Wang, Al-Saffar, Rogers, and Hahn Science Advances 2023? While these additional analyses are not necessary, they are straightforward and would add a lot to the paper, as the authors are in a unique position to do them well given the richness of the dataset.

5. Since it is very difficult for outside researchers to apply for access to the data that the authors used in this study, readers will have limited ability to either try to reproduce these results or to build on them along the lines of the above suggestions in point 4. The work the authors have put into calling mutations will likely have a greater impact if some limited individual-level data are released along with publication of the paper, for example, mutation counts and parental ages for each child in the study or ideally the counts of the 7 mutation categories that are in use (C>G, C>A, CpG>TpG, etc).

6. Materials and Methods, lines 274 - 289: Even though the DNM calls and phasing were performed earlier and described in a previous manuscript (Kaplanis et al, 2022), some basic analysis of their soundness should be presented, especially given that the final set of analysed DNMs may be very sensitive to these choices and that the aims of the previous paper were different to the ones in the present manuscript. A simple but effective check would be to compare the age-adjusted counts and spectra to those found in earlier papers like ref. 53, Jonsson et al. 2017, and ref. 31, Wang et al, 2023. Is there any indication that population differences in mapping quality and/or heterozygosity could be creating differences in DNM call quality among populations?

7. Materials and Methods line 336 and 369: Two of the linear models (those for trio DNM counts, line 336, and pyrimidine proportions, line 369) have a variable corresponding to mean parental age, but there is evidence that age has differential effects on both the rate and spectra of DNMs in mothers and fathers (see ref. 53, Jonsson et al. 2017, ref. 31, Wang et al,

2023 and ref. 52, Gao et al. 2019). It would be more appropriate to test for the effects of maternal and paternal age separately.

8. Materials and Methods, line 369 - 375: The quasi-binomial model does not seem appropriate to use here since separately estimated binary logistic regression models do not account for the compositional interdependence of the proportions of different mutation types. The authors could avoid this problem by estimating a single multinomial logistic regression model for each ancestral group, using the different substitution types as the multinomial categories.

9. Line 878: small typo, 1 should be x

10. Line 141 and 585: Supplementary Figure 4 and Figure 2 should be swapped because the supplementary figure seems likely to be of broader interest: readers will probably be most interested in how tobacco exposure influences overall DNM rates, and the breakdown by parent-of-origin is an interesting but more specific detail.

11. Line 466, 150- 159: Is there a reason why the heritability analyses weren't performed on ancestries other than EUR? A brief comment or explanation would be helpful, and the main text should discuss how the finding of zero heritability can be reconciled with the existence of differences between continental groups.

12. Lines 174-190: Some of the 'exposures' tested in the MR study did not seem to satisfy the requirements of MR analyses – for example, age of menopause can't have a direct effect on a child's DNMs since a child's birth always precedes their mother's menopause – this seems to be a case where there are one or more additional causal variables affecting both menopause and maternal germline mutagenesis. The same could be true of body mass index if it is measured long after the birth of the children and is affected by post-childbirth maternal exposures. Some discussion of this in either the main text or the Materials and Methods would be valuable.

Reviewer #3

(Remarks to the Author)

Reviewer #4

(Remarks to the Author)

This manuscript reports on differences in mutation rate and spectra among continental ancestry groups. The authors have also analyzed potential effects of epidemiological and biological covariates. Existing studies point to homogeneity of mutation rate in humans (in contrast to recombination rate). In this light, the analysis of the large-scale trio collection assembled by GEL is of high importance irrespectively of whether the result is positive or negative. The authors report several significant differences between populations. I also find the effect of smoking on germline mutation rate of great interest. I have several comments/suggestions listed below.

1) The authors employ a parametric model (with the logit link) to compare ancestry groups. I worry that deviations from the model may cause a spurious signal (especially with the highly imbalanced datasets). The study would benefit from demonstrating that the model is accurate. For example, the authors may sample from the model and demonstrate that the sampling produces counts identically distributed to the data. Another (or additional) possibility would be to employ a permutation-based strategy.

2) The statistical model employed by the authors uses a single parameter for the mean of the parental ages. This choice is hard to justify. It is plausible that age differences between parents are ancestry dependent. It is also known that maternal and paternal mutations have different distributions along the genome and different spectra. It is very possible that these effects are minor, but it would be better to see a model with maternal and paternal ages being separate covariates.

3) It would be of great interest to check if the spatial distributions of mutation along the genome (for example, clustered maternal mutations) show any differences by ancestry. This may shed some light on the underlying biology. I certainly appreciate that power can be an issue here.

4) It is of interest whether the apparent ancestry differences may be due to early somatic mutations in parents rather than mutations in the germline. This is a subset of all mutations called as de novos. Is it possible to see if the deviating classes of mutation calls have any notable shifts in coverage in parents?

5) I find the discussion of Harris and Pritchard, 2017 results somewhat misleading. I believe that there is a consensus in the community that the subtle SFS spike is due to a transient change in rate of specific mutations that that was caused by a factor (more likely environmental and less likely genetic) that is not active at present. It has nothing to do with the present-day differences presented in this manuscript.

Version 1:

Reviewer comments:

Reviewer #1

(Remarks to the Author)

I thank the authors for their thorough response to the reviews. I have no further remarks.

Reviewer #2

(Remarks to the Author)

The authors have done a very thorough job with revisions, and the manuscript is much improved. I have no further comments.

Reviewer #3

(Remarks to the Author)

Reviewer #4

(Remarks to the Author)

The revised manuscript is an improvement. I am satisfied with the authors' response to all original comments. As a result of re-reading the manuscript, I have a new quick comment (see below). I apologize that it was omitted from the original review (unless I am missing this).

1) The manuscript states that the rate of de novo mutations in parent-child trios of African ancestry is slightly elevated compared to trios of other continental ancestry groups. The main source of false-positive mutation calls is undercalling very rare heterozygous alleles in parents. African ancestry individuals have more rare heterozygous alleles per genome. It would be great to see evidence that this effect is not driving the signal.

Version 2:

Reviewer comments:

Reviewer #4

(Remarks to the Author)

I appreciate the quick response to my remaining comment. I apologize again for this comment surfacing so late in the process (especially having been on the receiving end of such late comments). Saying that, I do not find the response convincing. There is no indication that calling per family rather than per batch solves the problem. First, individuals of African ancestry have more heterozygous sites. If there is a small fraction of false-positive de novo mutation calls due to undercalled hets in parents, it would masquerade as the difference in rate of de novo mutations between ancestry groups. Second, in many variant calling methods, allele frequency information indirectly leaks into family-based calls. We have never tested Platypus, but the error profile of trio-based calls using other methods (e.g. GATK) suggests that they remain sensitive to population allele frequency. The observed effect is subtle and is the main result of the paper. It would be helpful to make sure that it does not result of a variant calling artifact.

Depending on the stringency of the calling procedure, it may generate either more false-positive calls in families of African ancestry or more false-negative calls in families of other ancestries. It would be very helpful to obtain an upper bound estimate of the error rate by ancestry. If the authors have access to multi-generational families, it may provide the best way to analyze the errors due to undercalled hets. Otherwise, testing the pipeline on gold standard datasets and plotting error rates by coverage and allele frequency would be helpful.

Version 3:

Reviewer comments:

Reviewer #4

(Remarks to the Author)

The analysis of VAF and inclusion of heterozygosity in the model do suggest that the effect is either absent or minor. I do not have other concerns. The authors may choose to show that the difference between populations persist even after the exclusion of all ALT==1 sites in parents.

To respond to the authors' question about multigenerational data, there is a multigenerational public dataset of a large Utah family. It does not allow for the comparisons between ancestry groups but does allow for estimation of the number of over/undercalled de novo mutations.

The impact of ancestral, environmental and genetic influences on germline de novo mutation rates and spectra

We thank the reviewers for their insightful comments and valuable suggestions. Their feedback has significantly improved the clarity of our manuscript and strengthened the robustness of our findings. Below we have addressed each of the reviewers' points, **in blue**. Corresponding revisions made in the main manuscript are marked **in green** for ease of reference.

Following (or in addition) to the reviewers suggestions, we have implemented five major changes in the paper:

1. In response to **comments 2.7** and **4.2**, we are controlling for both parental ages (maternal and paternal age at conception) in our DNM counts and spectra regression models. Below, we include a comparison of the results prior and after implementing this change in our DNM counts to ancestry regression (**Model 1**). This rendered similar results to those we presented originally (**Rebuttal Figure 1**), with the exception of the EUR and SAS pair, where differences in DNM counts are no longer significant.

Rebuttal Figure 1. Ancestry association to DNM counts prior and after modifying Model 1.

Ancestry effect sizes and directions using the original Model 1 (red), including a single covariate for parental ages at conception (i.e. mean parental age), and a modification of Model 1 (purple), including paternal and maternal ages at conception as separate covariates. Effect direction corresponds to the first ancestry in each row pair (e.g in the first row, DNM counts for the AFR group is, on average, n times higher than that of the AMR group). Asterisks indicate significance after multi-testing correction (FDR 5%).

2. In addition to including both parental ages as covariates, and **following comment 2.8**, we have modified our modelling strategy for DNM spectra (**Model 2**), to account for the interdependence between pyrimidine substitution proportions. We are now implementing the compositional data modelling strategy described in by Zhou et al., 2022 ¹. Briefly, this method (linDA) models centred log-ratio transformed proportions using linear regression and applies bias correction to effect estimates and p-values to account for the compositional nature of data, where components are interdependent and constrained to sum to a constant ¹. After applying the aforementioned modifications, we only detected significant associations for C>A and CpG>TpG levels between the EUR and SAS (5% FDR). To test how many of our previous associations were likely due to systematic differences at age at conception between parents, we repeated our compositional analysis using our original **Model 2** formula (i.e. mean parental ages only), with which we were able to retrieve the same associations between the EUR and SAS pairs at 5% FDR (**Rebuttal Figure 4**). For completeness, we also re-ran these models using the pseudo-binomial regression method to compare how many of the changes to our original results were attributable to the compositional bias correction implemented by linDA and/or systematic differences in parental age at conception. We found our previously reported increased T>C proportions for the EUR group as compared to the SAS group (**Rebuttal Figure 4**) were likely attributable to compositional bias, while our previously reported C>G differences were likely driven by both age differences and compositional bias.
3. We have improved clarity on our **main Figure 1**, and have replaced it in the new draft. This can be also found in our response to **comment 2.3**.
4. In response to comment **comment 2.4**, we have extended our **Supplementary Note #2** to include a new analysis to demonstrate that cross-ancestral differences are not driven by age differences.
5. We have corrected an error related to the identification of CpG sites. This occurred because the strand direction (i.e. 5' => 3') of CpG pairs was not considered, leading to an overestimation of CpG>TpG sites. We have now corrected the pyrimidine substitution proportions per trio. The revised data has been used to address further reviewer's comments.

Reviewer #1 (Remarks to the Author)

DNA mutation rate is a quantitative trait that can vary between cells and individuals, and, and it is understood that this trait is influenced by genetic variation in multiple species. In this study by Garcia-Salinas, et al, use one of the largest datasets of germline mutation assembled to investigate environmental and genetic influences on mutation rate in humans. The authors find associations of ancestry, smoking, and age of menopause on de novo mutation rate. Overall I find the manuscript to be well written and the analyses to be rigorous and thorough. I only have minor comments.

We thank the reviewer for these positive comments.

Comments:

1.1 Could the authors please describe in the methods whether the sex chromosomes and mitochondrial genome were included in this analysis? That information may be contained in Kaplanis et al 2022 but it would be helpful for the reader not to have to revisit that.

We only included *de novo* SNVs from the autosomes and chromosome X when generating per trio counts. However, variants from the mitochondria and Y chromosomes were excluded from the analysis. For analyses of parentally phased DNMs, we excluded chromosome X, as phasing was performed exclusively for the autosomes. We have now added this in the methods section (lines 348-349), so readers do not need to refer back to Kaplanis et al 2022.

Lines 348-349: *“This dataset included only mutations from autosomes (chr1-22) and chrX, but did not include DNMs from chrY and mitochondria”*

1.2 I found references to model 1,4,5,6,7, and 8. But there doesn't seem to be a model 2 or model 3?

Thank you for bringing this to our attention. **“Model 2”** and **“Model 3”** were part of earlier versions of the manuscript. We have now re-enumerated the equations to accurately reflect the models in the current version of the paper.

1.3 The author use a number of technical and biological covariates in their mutation models (i.e. Models and 4-8 referenced above). Can they briefly explain why these covariates were selected, and whether there is any statistical evidence that each of these covariates influences DNM count? In other words, I'm looking for justification of the inclusion of the “nuisance” parameters so to speak, not the main effect (hypothesis-driven) parameters (like “ever smoked”). If this has already been explored in Kaplanis et al 2022 that is fine, but it would be good to explicitly state this in the methods (i.e. that each of these terms has been demonstrated to influence total observed DNM).

We included these nuisance covariates as they were found to explain at least 1.3% of the variance in the GEL DNM counts reported in Kaplanis et al. ². This is now clarified in lines

395-402. Having used a different generalised linear model (quasi-Poisson) and a cohort subset from that originally reported by Kaplanis et al. ², we recalculated variance explained by nuisance covariates with partial R², applying the variance-function method from the *rsq.partial()* function (R package *rsq*). With this, we estimate that technical covariates account for ~1.68% of DNM count variance in this cohort (**Rebuttal Table 1**). While this value is similar to Kaplanis' original estimate, we note that the partial R² may underestimate total variance by not fully capturing complex interactions or shared variance among covariates. Additionally, covariates such as parental coverage and read alignment percentages explained minimal variance, and removing them had a negligible effect on ancestry estimates (**Rebuttal Figure 2**). Nevertheless, to maintain consistency with previous publication, we retained these covariates

Covariate	Partial R² (percent)
Trio ancestry	0.16
Age at offspring mother	3.58
Age at offspring father	28.12
Illumina mean coverage Mother	0.018
Illumina mean coverage Father	0.039
Illumina mean coverage Offspring	1.18
Percent of aligned reads Mother	0.00024
Percent of aligned reads Father	0.00108
Percent of aligned reads Offspring	0.239
Trio median Bayes factor	0.171
Trio median VAF	0.028

Rebuttal Table 1. Variance explained (partial R²) per covariate in model 1

Rebuttal Figure 2. Ancestry effect estimates from Model 1 versions keeping (blue) or discarding (pink) parental sequencing stats as extra covariates.

Lines 395-402: *“Apart from these biological factors, technical variables associated with sequencing quality metrics in the family trio have been shown to explain up to 1.3% of the variance of DNM counts in this cohort 4. These metrics include: the mean sequencing depth (per trio member), the percent of aligned reads (per trio member), as well as trio level summary statistics such as the median Bayes factor (from the Platypus variant caller) and the median variant allele fraction (VAF) of all DNMs called in a given trio 4. All of these biological and technical covariates are accounted for in our models.”*

1.4 Line 101 “The differences in C>A and T>C proportions recapitulate what has been previously reported [14]). I don’t think this sentence fully captures what is going on here and led me to spend some time on Harris and Pritchard until I read the Supplementary Note 3. I would recommend something like: “The differences in C>A and T>C proportions seen between EUR and SAS are recapitulated using a different data set and a different method for quantifying population-specific germline mutation spectra [14]).”

The reviewer makes a good point. We apologise for the vagueness of the original sentence. We have reworded according to the reviewer’s suggestion (and our findings after addressing further peer review comments). This change can be found in **lines 110-113**.

Lines 110-113: *“The differences in C>A and CpG>TpG proportions observed between EUR and SAS groups recapitulate those seen using a different data set and a different method for quantifying population-specific germline mutation spectra based on polymorphism data.”*

1.5 Line 246 the authors sound a call to collect “comprehensive epidemiological data on potential environmental mutagens”. To that end, can the authors describe what type of information is available on the geographic sampling location of the subjects in this study? Is there any sort of GPS encoding of the subject’s present day or historic domicile(s)?

We thank the reviewer for this question. The Genomics England dataset includes access to the National Health Service (NHS) electronic health records, collected during hospital and general practice visits from participants in the 100,000 Genomes Project ³. While detailed residential addresses are not available, broad demographic and geographic metrics, such as NHS commissioning regions and Multiple Deprivation Indices (IMD) are available. NHS commissioning regions are administrative areas responsible for delivering healthcare services, and the IMD provides socioeconomic rankings based on factors like income, employment, and health. Though these metrics could be potentially useful for approximating exposure to environmental mutagens, using these data in epidemiological analyses would require significant additional data curation and integration. Even after this, it is not guaranteed that any specific composite metric will reveal a pattern that can be easily interpreted in terms of DNM mutation counts and spectra. We recognise the value of this suggestion and plan to explore it further in future research. However, we foresee that this question will be better addressed with the advent of DNM datasets that specifically collect life-history information on the participants (e.g. number of years since smoking regularly, and occupation before conception). We have edited our discussion to incorporate this.

Lines 304-308: “This work demonstrates the imperative to include greater diversity of genetic ancestry in DNM studies, and to collect comprehensive epidemiological data on potential environmental mutagens, coupled with life-history information on participants. This will enable more powerful and robust investigation of the genetic and environmental influences on germline mutation”.

1.6 Line 539 the URL is incorrect:

(<https://www.genomicsengland.cgfbo.uk/research/academic/join-gecip> [[genomicsengland.cgfbo.uk](https://www.genomicsengland.cgfbo.uk)]) should be “co.uk” not cgfbo.uk

We thank the reviewer for pointing out this and have now corrected the error accordingly. This change can be found in **line 639**.

1.7 Supplementary Figure 8: The legend could be clarified a bit. I believe this figure is a cartoon showing properties of three different distributions plotted against simulated data (grey circles). What type of data are the grey circles meant to represent? Why are some circles larger than others? I assume the symbolism is meant to be analogous to the binned mutation data shown in Supp Fig 9A and 9B. But I don’t want to assume too much.

We thank the reviewer for this suggestion. We have reworked the footnote of the (now) **Supplementary Figure 12**. This change can be found in **lines 771-777**, and we are also including this below:

Lines 773-779: “Illustration of the mean-to-variance relationship of the quasi-poisson distribution (blue line, b). Each circle represents N theoretical samples grouped after binning a continuous variable (e.g. parental age), with radius proportional to N, and are placed in the

graph according to the mean and variance of their respective samples. The overall distribution of this theoretical data follows a quasi-Poisson distribution due to the mean-variance overdispersion (i.e. variance exceeds the mean in a linear manner). For contrast, the Poisson (yellow line, a) and negative binomial (red line, c) distributions are also shown”.

Reviewer #2 (Remarks to the Author)

De novo mutation (DNM) rates are an important 'meta' phenotype because of their influence on many other biological processes, including evolution, disease and ageing. However, as the authors of this manuscript note, it's unusually difficult to study DNMs and their associated properties, because detecting them requires genomic information from parents of the samples in addition to the samples themselves (i.e. trio data).

The authors of this manuscript take advantage of a large dataset of trios and DNM calls previously published by Kaplanis et al, 2022 to conduct a variety of biostatistical analyses about DNM rates. The direct associations between tobacco exposure and DNM rates presented in this manuscript are entirely novel, as are some of the results about DNM rate and spectrum differences between people of different continental ancestries. Because the dataset is considerably larger than those in comparable studies, the authors are also able to readdress several long-standing scientific questions that previous researchers had insufficient power to answer. For instance, a prior SNP heritability analysis from Kessler et al. 2020, cited by these authors, uses around ~90 K DNMs (nearly 10 times fewer than the present work) from around 1500 trios (~5 times fewer than the present work).

The methodology is largely sound and thoroughly described, though some figures have ambiguous interpretations and should be redesigned (see “minor comments” for details). The conclusions are generally justified and appropriately cautious. Given its combination of novel results and timely reanalyses, we think this manuscript usefully pushes forward the state of knowledge on causes of DNM rate and spectrum variation within the general population.

We thank the reviewers for these positive comments and hope that our revised figures and manuscript meet with their approval.

2.1 The content needs to be restructured to make the overall scientific narrative clearer. At present, the manuscript presents many different analyses in sequence without much commentary on how they relate to each other. The clarity and likely impact could be increased by re-organising the “Results” to reflect the three different factors in the title (ancestral, environmental and genetic influences on DNM rates). A section on “ancestral influences” could cover all the results about DNM rates and spectra between different continental groups. These results motivate the rest of the manuscript: the observation that there *are* systematic differences in DNM rates and spectra across both families and ancestries indicates there may be genetic and environmental influences on DNM rates that we are yet to find. A section on “genetic influences” could cover the SNP heritability results and GWAS results, and a section on “environmental influences” could cover the smoking and Mendelian randomisation analysis.

We have now restructured the paper to clarify the scientific narrative, following the reviewer's suggestion.

2.2 After re-organising the “Discussion” section to match this structure, it would also be useful to explicitly mention the ways in which the present study was not able to directly address some of the questions posed in the introduction. For instance, the low SNP heritability in EUR samples doesn't necessarily mean there aren't genetic influences on DNM rates that act between ancestral groups (or within the many ancestral groups that weren't tested in the present study). In addition, the authors should clearly point out that uneven sample sizes between continental groups likely has a large impact on which pairs of populations are found to have significantly different mutation rates and spectra.

We agree with these points and have reorganised the main text and our discussion section to reflect this change. We have also added the following specifications to point out the limitations/reasoning of our analyses:

Lines 142-146: *“Ancestry differences in DNM rates and spectra could reflect genetic differences between ancestry groups²¹ that influence the germline mutation rate. Since detecting causal ancestry-stratified variants is likely to be difficult, we instead explored the contribution of common genetic variants to DNM rate within the largest ancestral group within our cohort, restricting to unrelated parents inferred to have European genetic ancestry(...).”*

Lines 155-156: *“In spite of this finding, we note that potential effects of ancestry-associated genetic variation on DNM rate may not be detected by this analysis”.*

Lines 248-257: *“Furthermore, our ability to detect differences in DNM rate and spectra across ancestry pairs may have been greatly impacted by the sample size differences between groups in our cohort.”*

Lines 284-286: *“Further, larger sample sizes of diverse ancestral backgrounds will be needed to directly address whether ancestry-associated genetic variation significantly influences DNM rate”.*

Lines 308-314: *“Although based on a larger cohort than previous studies, our analyses were nevertheless limited in their power to detect the effects of rare genetic variants, or more subtle differences across ancestries and sexes on DNM rate variation. Since existing methods for detecting and estimating genetic associations rely on genetically homogeneous cohorts, future studies may also require methodological developments to effectively disentangle ancestry-correlated genetic variants from environmental factors affecting de novo mutation rates and patterns”.*

2.3. Figures 1A and 1B are very difficult to parse: the reader has to look both across a row and up a column to see all the cells relevant to a certain continental population, and they then have to think about how the model reference changes across these cells to interpret the numbers inside them. It should be made more obvious which group is elevated with respect to which other group—for example, does an “excess”

mean the “row” group is elevated over the “column” group or vice versa? We suggest re-running these analyses using a common model reference in each case (‘EUR’, perhaps), and then presenting the results with a barchart instead of a heatmap (similar to Supplementary Figure 2). This presentation would also allow confidence intervals to be displayed, and so would help the reader get a sense of which results might fall below statistical significance simply because of sample size (those involving the EAS population, for instance).

We thank the reviewer for their useful suggestions. We have modified the **Main Figure 1** to be more informative about effect units and directions regarding ancestry. Changes include showing the baseline number of DNMs expected for each ancestry after controlling for technical and biological covariates (Main **Figure 1A**), and showing fold change differences in pyrimidine substitution proportions between ancestry pairs (Main **Figure 1B**). For completeness, we are also including new supplementary material, showing all non-redundant ancestry pairwise comparisons for both DNM counts and pyrimidine substitution proportions (Supplementary **Figures 1** and **2**, respectively). These plots represent the fold change associated with a given ancestry (first on row) as compared to another (second on row), for all the possible non-redundant ancestry pairs (n=10). Point estimates include 95% confidence intervals, and delineate significant associations after FDR correction. The new **Main Figure 1** is shown below:

Lines Figure 122-130: Ancestry associations to DNM rate and DNM spectra . A) Baseline DNM counts expected for each ancestry group after controlling for technical and biological covariates. Bars represent 95% confidence intervals and asterisks indicate significant differences between groups (FDR<5%). B) Pyrimidine substitution proportion fold changes (log2 scale) for all possible pairwise ancestry group comparisons across 7 pyrimidine substitution categories (n=70). Ancestry pairs with significant fold change differences after adjusting for multiple testing (FDR<5% - red line, FDR<10% yellow line) are labelled. Effect estimates correspond to the first ancestry in each label pair (e.g. proportions of C>A in the EUR group being on average 0.98X smaller than those of the SAS group).

2.4 The finding of significant mutation spectrum differences between human continental groups is one of the most interesting findings of the paper given that this has been long suspected based on segregating variation but was not detectable using previous smaller DNM datasets. Given this, the paper could benefit from additional efforts to interpret these differences.

- **Do the authors think that the finding of no significant differences between AFR and other groups is mainly driven by the small sample size of the African trios, or is the sample size sufficient to rule out the magnitude of effects predicted by the Harris and Pritchard study?**

We thank the reviewer for their useful comments. In response to the question about whether the lack of significant DNM spectra associations in ancestral groups with smaller representation compared to the SAS and EUR groups, and as compared to the polymorphism data from Harris and Pritchard, 2017 ⁴, could be due to smaller sample sizes, we conducted additional analyses to assess this possibility.

First, we re-evaluated the significant DNM spectra differences detected in our analysis ($FDR \leq 5\%$). We specifically focused on C>A and CpG>TpG substitutions between EUR and SAS groups, for which we observed a mean fold change of 0.93 (i.e. 6.51% absolute change). To investigate the impact of sample size on our ability to detect a significant effect, we performed a downsampling experiment by randomly subsampling the SAS group ($n = 1,250$) to sizes of 198 (the sample size of the AFR group in our data), 250, 500, and 800 trios. For each subset, we repeated the DNM spectra associations using **Model 2**, performing 1,000 iterations at each sample size. We then averaged the log2 fold changes and standard errors across these iterations for each pyrimidine substitution. Our results showed that downsampling the SAS group to 198 trios (matching the sample size of the AFR group) did not produce any significant fold change estimates for C>A or CpG>TpG substitutions (i.e. the fold changes were not significantly different from the null expectation of 1). Furthermore, we determined that a significant fold change for these substitution types was only observed with a sample size >250 SAS trios (**Rebuttal Figure 3**).

Rebuttal Figure 3. Fold change estimates (EUR/SAS) for C>A and CpG>TpG levels.

Fold changes and standard errors represent the mean across 1000 random subsamples from the SAS population. Subsample size is indicated in the y-axis. The top bar corresponds to the regression on the original SAS sample size (n=1,250). Asterisks correspond to significant differences against the null hypothesis of no fold change difference ($p \leq 0.05$).

We then looked at the effect sizes observed in polymorphism data from Harris and Pritchard, (Supplementary table 5). In this data, the strongest fold change is 0.956 (i.e. 4.3% absolute change) for C>T substitutions between the equivalent EAS and EUR groups in that study. Taking into account that in our DNM data a sample size of >250 SAS trios was necessary to detect a significant fold change that is higher than the strongest effect observed in polymorphism data (i.e. ~4.3% change), this suggests that if the mutagenic factors affecting polymorphisms are still active to the same extent in contemporary populations, we may have failed to detect them due to the sample sizes of particular ancestries (n AFR = 198, n AMR = 215, n EAS = 53). Note however that a formal power analysis would require simulation with multiple mutational and ancestry-specific parameters that are currently unknown.

We have included this analysis in Supplementary Note 4 and Supplementary Figure 19 in our new paper draft.

- **Are the observed differences among the mutation spectra of continental groups explainable by differences in the distribution of parental ages, as recently suggested by Wang, Al-Saffar, Rogers, and Hahn Science Advances 2023? While these additional analyses are not necessary, they are straightforward and would add a lot to the paper, as the authors are in a unique position to do them well given the richness of the dataset.**

We have now performed a careful revision on parental age differences across ancestries, and we have applied different strategies to test whether our observations could be driven by such differences. We'd like to note that, by reviewers suggestion, we have now modified **Model 2** to include parental ages at conception as separate covariates rather than including them as a single covariate (i.e. mean parental age) (Rebuttal Figure 4). We believe that including a covariate term for both the paternal and maternal ages as independent covariates will remove any spurious associations between ancestry and DNM rate/spectra that are due purely due to ancestral differences in parental ages at conception alone.

Rebuttal Figure 4. Ancestry association to DNM spectra prior and after modifying Model 2.

A) Across all non-redundant ancestry pairs (n=10), and **B)** for the EUR SAS pair only. Ancestry effect sizes and directions using the original Model 1 (red), including a single covariate for parental ages at conception (i.e. mean parental age), and a modification of Model 1 (purple), including paternal and maternal ages at conception as separate covariates. Shades correspond to regression methods used: linDA in darker colours and pseudo-binomial in light colours. Effect direction corresponds to the first ancestry in each row pair (e.g in the first row, DNM counts for the AFR group is, on average, n times higher than that of the AMR group). Asterisks indicate significance after multi-testing correction (FDR 5%).

To be fully exhaustive on ensuring that our observations are not due to systematic differences between parental ages at conception across ancestries, we performed the subsampling and bootstrapping analyses described in the **Supplementary Note 2**. Here, we matched paternal or maternal ages at conception across ancestries according to whether given ancestry pairs had significant parental age differences and significant differences in DNM counts or spectra. This experiment revealed that our original observations are maintained after subsampling and age matching.

2.5 Since it is very difficult for outside researchers to apply for access to the data that the authors used in this study, readers will have limited ability to either try to reproduce these results or to build on them along the lines of the above suggestions in point 4. The work the authors have put into calling mutations will likely have a

greater impact if some limited individual-level data are released along with publication of the paper, for example, mutation counts and parental ages for each child in the study or ideally the counts of the 7 mutation categories that are in use (C>G, C>A, CpG>TpG, etc).

We agree that it would be ideal to release DNM counts so that our work is more easily reproducible. We have asked Genomics England about this possibility but we did not get a positive response due to their strict data protection policies. In particular, their policy precludes the release of data or summaries for categories comprising five or fewer individuals. Since this includes many of the parental age combinations in our dataset, we would have to either exclude categories or bin the data by age, making it considerably less useful. We note that Genomics England is open for new researchers accessing this data inside their research environment; see <https://research.genomicsengland.co.uk/application/>. We have prepared a folder of key files and code to be able to reproduce our analyses at /re_gecip/sharedAllGeCIPs/aeg_dnm_2024/.

2.6 Materials and Methods, lines 274 - 289: Even though the DNM calls and phasing were performed earlier and described in a previous manuscript (Kaplanis et al, 2022), some basic analysis of their soundness should be presented, especially given that the final set of analysed DNMs may be very sensitive to these choices and that the aims of the previous paper were different to the ones in the present manuscript. A simple but effective check would be to compare the age-adjusted counts and spectra to those found in earlier papers like ref. 53, Jonsson et al. 2017, and ref. 31, Wang et al, 2023. Is there any indication that population differences in mapping quality and/or heterozygosity could be creating differences in DNM call quality among populations?

We thank the reviewer for their suggestions. To address this, we compared the Genomics England (GEL) and deCODE project DNM callsets ⁵. We chose deCODE for this comparison as this sequenced *de novo* mutations directly, while Wang et al., 2023 only inferred *de novo* mutations from polymorphism data ⁶. To compare against our GEL dataset, we first filtered the deCODE DNM dataset to *de novo* SNVs only, which we then classified into 7 possible pyrimidine substitutions (i.e. C>A, C>T, CpG>TpG, C>G, T>A, T>C, T>G). As in GEL, we obtained the proportion of DNMs in each pyrimidine substitution class per trio.

We combined the deCODE data with the European-ancestry participants from GEL (n=8,104), then regressed DNM counts and pyrimidine substitution proportions on parental ages alone (i.e. maternal + paternal age at conception) using the same kinds of models used for our main analyses: generalised linear regression (quasipoisson family) for DNM counts, and a compositional linear regression for pyrimidine substitution proportions. We extracted the residuals for each regression and compared them between datasets using the Wilcoxon rank sum test. While controlling for parental ages alone, we found significant differences between the datasets in the residual DNM counts and the residual proportions of all seven pyrimidine substitutions (**Rebuttal Figure 5** and **6**). We assume this is due to batch effects across datasets.

Potential sources of these include sequencing depth (GEL: average 32X, DECODE: average 35X), sequencing technology (GEL: HiSeq 2500, DECODE: GAIIx, HiSeq 2000/2500, or HiSeq X instruments); *de novo* mutation calling algorithm (GEL: Platypus + GEL *de novo*

filtering, DECODE GATK-unified genotype caller + DECODE *de novo* filtering)^{5,7}. If we had individual-level QC metrics for the deCODE dataset, we could control for these in the same way we have for GEL, and potentially reduce these batch effects. However, for the reasons outlined below, we think it is unlikely these technical factors are driving our results.

First, reassuringly, we found that the parental age effects on DNM counts in the deCODE dataset were indistinguishable from those in our GEL dataset (**Rebuttal Table 2, Rebuttal Figure 7**). Secondly, our models account for the relevant technical covariates. Finally, regarding mapping differences across ancestries, we assessed this in our original submission using the methods described in our **Supplementary Note 2** (See the section - “*Ensuring that the associations between ancestry and DNM rate are not due to technical artefacts or ascertainment bias*” section). Briefly, we first retrieved the reads mapping to the alternative allele from each reported *de novo* mutation per offspring. We then calculated the average number of mismatches (NM) from these reads per DNM, and then we calculated 2 metrics: the average average NM per trio, and the maximum average NM per trio. We performed pairwise comparisons of these metrics across all ancestries (**Supplementary Figure 3**), but none of these showed significant differences for either metric. Finally, we included each NM metric as an extra covariate in our Model 1 regression, and found very similar results to our original models (**Supplementary Figure 3**).

Rebuttal Figure 5. Dataset pairwise comparison of parental age residualised DNM counts.

Showing residuals extracted from a model accounting for parental ages alone (left), and a model including parental ages plus a dummy covariate indicating the dataset origin (GEL | DECODE), Asterisks indicate significant differences (Wilcoxon rank-sum test $p \leq 0.05$). NS - Non-significant differences (Wilcoxon rank-sum test $p > 0.05$). Dotted red line intersects zero in the y-axis.

Rebuttal Figure 6. Dataset pairwise comparison of parental age of residualised DNM spectra.

Showing residuals extracted from a model accounting for parental ages alone (A), and a model including parental ages plus a dummy covariate indicating the dataset origin (GEL | DECODE - B). Asterisks indicate significant differences (Wilcoxon rank-sum test $p \leq 0.05$). NS - Non-significant differences (Wilcoxon rank-sum test $p > 0.05$). Dotted red line intersects zero in the y-axis.

Rebuttal Figure 7. Dataset pairwise comparison of parental age effects on DNM counts per parental year of age.

Dataset	Parent	# DNMs per year	95% confidence interval
GEL	Father	1.24	1.20-1.28
GEL	Mother	0.43	0.38-0.48
deCODE	Father	1.22	1.14-1.30
deCODE	Mother	0.48	0.37-0.59

Rebuttal Table 2. Estimated DNM counts per year of parental age in either fathers and mothers.

2.7 Materials and Methods line 336 and 369: Two of the linear models (those for trio DNM counts, line 336, and pyrimidine proportions, line 369) have a variable corresponding to mean parental age, but there is evidence that age has differential effects on both the rate and spectra of DNMs in mothers and fathers (see ref. 53, Jonsson et al. 2017, ref. 31, Wang et al, 2023 and ref. 52, Gao et al. 2019). It would be more appropriate to test for the effects of maternal and paternal age separately.

We thank the reviewer for their useful suggestion. We have addressed this by including a separate parental age at conception for each parent in **Model 1** and **Model 2**. We are including a comparison between the results of our previous models and their updated versions at the beginning of this document and in our response to **comment 2.4 (Rebuttal Figure 1 and 3)**.

2.8 Materials and Methods, line 369 - 375: The quasi-binomial model does not seem appropriate to use here since separately estimated binary logistic regression models do not account for the compositional interdependence of the proportions of different mutation types. The authors could avoid this problem by estimating a single multinomial logistic regression model for each ancestral group, using the different substitution types as the multinomial categories.

We agree that the quasibinomial model was not the most appropriate approach in this case as this does not account for the multivariate nature of compositional data, as the reviewer points out. To address this issue, we are now modelling proportions using the compositional strategy described in by Zhou et al., 2022¹. We are including a comparison between the results of our previous models and their updated versions in our response to **comment 2.4 (Rebuttal Figure 4)**.

2.9 Line 878: small typo, 1 should be x

We thank the reviewer for finding this. The typo was at " $S_{p1(m)}$ " in **Supplementary Note 3**, we have now corrected this.

2.10 Line 141 and 585: Supplementary Figure 4 and Figure 2 should be swapped because the supplementary figure seems likely to be of broader interest: readers will probably be most interested in how tobacco exposure influences overall DNM rates, and the breakdown by parent-of-origin is an interesting but more specific detail.

We thank the reviewer for this suggestion. However, we have not swapped the figures because we believe that the results on phased DNMs are more accurate than those on trio DNMs. This is because the electronic health record data on smoking is per individual, and is incomplete for many trios, so the trio results include a trio-level proxy to smoking behaviour, which introduces noise (see **Methods - Analysis of smoking and DNM rate** section). On the other hand, the associations ran using phased DNMs (our current **Main Figure 3**) include only individuals with available health records, and allows us to directly test the smoking effect on the phenotype-matched individual (i.e. the parent and their respective germline mutation rate).

2.11 Line 466, 150- 159: Is there a reason why the heritability analyses weren't performed on ancestries other than EUR? A brief comment or explanation would be helpful, and the main text should discuss how the finding of zero heritability can be reconciled with the existence of differences between continental groups.

Our choice was based on sample size, due to the EUR group being far better represented in the GEL cohort than any other ancestry. We have clarified the reasoning behind this choice in the **lines 140-146**. We have also clarified that our heritability analysis may not detect potential ancestry associated genetic effects in the **lines 151-154**. Regarding the question: **“how the finding of zero heritability can be reconciled with the existence of differences between continental groups”**, we have reorganised our discussion section to give our views on the sources of this discrepancy and how these may be addressed in future research efforts.

Lines 142-148: *“Ancestry differences in DNM rates and spectra could reflect genetic differences between ancestry groups²¹ that influence the germline mutation rate. Since detecting causal ancestry-stratified variants is likely to be difficult, we instead explored the contribution of common genetic variants to DNM rate within the largest ancestral group within our cohort, restricting to unrelated parents inferred to have European genetic ancestry (7,786 mothers, 7,692 fathers). We first estimated the variance in DNM rate explained by variants with minor allele frequency $\geq 0.1\%$, using GREML-LDMS”*

Lines 153-156: *“From this, we concluded that variance explained by common variants on DNM rate within this European-ancestry population must be too low to be detected in this sub-cohort. In spite of this finding, we note that potential effects of ancestry-associated genetic variation on DNM rate may not be detected by this analysis” .*

2.12 Lines 174-190: Some of the ‘exposures’ tested in the MR study did not seem to satisfy the requirements of MR analyses – for example, age of menopause can't have a direct effect on a child's DNMs since a child's birth always precedes their mother's menopause – this seems to be a case where there are one or more additional causal variables affecting both menopause and maternal germline mutagenesis. The same could be true of body mass index if it is measured long after the birth of the children and is affected by post-childbirth maternal exposures. Some discussion of this in either the main text or the Materials and Methods would be valuable.

We agree and thank the reviewer for bringing this to our attention. We have specified the reasoning behind the selection of these traits in the Methods section in **lines 617-623**.

Lines 619-625: *“We note that phenotypes such as age at menopause and BMI represent latent variables encompassing biological processes likely affecting the germline mutation, rather than affecting the germline mutation themselves. For example, age at menopause is thought to reflect the efficiency of the DNA repair machinery, which itself likely influences germline de novo mutation 12. In this sense, the strict MR assumption of no pleiotropic influence of the instruments on the outcome does not hold, and the measured effect reflects genetic correlation rather than causality”.*

Reviewer #3 (Remarks to the Author)

We thank the reviewer for their incisive review and hope they will be happy with our responses.

Reviewer #4 (Remarks to the Author)

This manuscript reports on differences in mutation rate and spectra among continental ancestry groups. The authors have also analysed potential effects of epidemiological and biological covariates. Existing studies point to homogeneity of mutation rate in humans (in contrast to recombination rate). In this light, the analysis of the large-scale trio collection assembled by GEL is of high importance irrespective of whether the result is positive or negative. The authors report several significant differences between populations. I also find the effect of smoking on germline mutation rate of great interest. I have several comments/suggestions listed below.

We thank the reviewer for their enthusiastic response to our manuscript.

Comments:

4.1 The authors employ a parametric model (with the logit link) to compare ancestry groups. I worry that deviations from the model may cause a spurious signal (especially with the highly imbalanced datasets). The study would benefit from demonstrating that the model is accurate. For example, the authors may sample from the model and demonstrate that the sampling produces counts identically distributed to the data. Another (or additional) possibility would be to employ a permutation-based strategy.

We thank the reviewer for their useful suggestions. We performed bootstrapped subsampling to test the accuracy of our model to predict DNM counts that resemble those of

the GEL trios. Additionally, we used bootstrapped permutation to estimate the likelihood of our model to return spurious ancestry association signals to DNM counts.

The first analysis was performed as follows. First, we randomly subsampled 80% of the cohort used to test the associations shown in the **main Figure 1** (n=9,820), and used this to train our DNM counts to ancestry regression model (i.e. **Model 1**). We then used this model to predict DNM counts in the remaining 20% of the cohort (i.e. the test dataset), and obtained the Pearson correlation between observed and predicted DNM counts for all of the ancestries altogether, and then for each ancestry separately. We repeated this process 2000 times. We observed that the mean Pearson correlation coefficients between predicted and observed DNMs ranged from 0.66 ($\pm 6e-3$, 95% confidence interval [CI]) to 0.73 ($\pm 5e-4$, 95% CI) across different ancestries (**Rebuttal Figure 8**). The lowest correlation was observed in the AMR samples, while the highest was in the EUR samples. Despite differences in sample sizes for the SAS and AFR groups (n trios = 1,250 and 198, respectively), our model performed similarly in predicting DNM counts in both groups, with a mean correlation coefficient of 0.684 for the AFR sample ($\pm 4.2e-3$, 95% CI) and 0.685 for the SAS sample ($\pm 1.4e-3$, 95% CI).

Rebuttal Figure 8. Mean Pearson's R between observed and predicted values in the test dataset across 2,000 random sub-samplings.

Pearson's R was calculated either for the whole cohort (i.e. "none" - no ancestry subset), or each ancestry separately. Error bars represent 95% confidence intervals for the mean estimate.

Next, we tested the likelihood of **Model 1** returning spurious associations. For this, we randomly permuted the trio-level ancestry labels in the whole test dataset (n = 9,820), each time running **Model 1** on the ancestry randomised dataset. We repeated this process 10,000 times, and counted the number of times each ancestry pair was significantly associated with DNM counts. For this, we counted the instances where an ancestry pair was associated with DNM counts with an FDR adjusted $p \leq 0.05$ in a single run of **Model 1**. We coded these

events as “PASS”, and the opposites as “FAIL”. We then compared “PASS”/“FAIL” counts per ancestry pair against the corresponding counts for the rest of ancestry pairs (i.e. aggregate counts across ancestry pairs) in a Fisher exact test to ask if a given ancestry pair was more likely to return spurious significant associations as compared to the background. We found that none of the ancestry pairs we report as having significantly different DNM counts in our main results are more likely to report spurious associations across 10,000 ancestry label permutations (Odds Ratio [OR] = 1, **Rebuttal Figure 9A**). Additionally, we found that the EUR and SAS pairs are significantly less likely to return significant associations with DNM counts as compared to the background (OR = 0.69, ± 0.15 95% CI, $p = 2.4 \times 10^{-3}$), presumably because they are the two biggest groups. Finally, we calculated the mean effect estimate and standard error per ancestry pair across all repetitions, from which we found no significant association to DNM counts for any ancestry pair across all randomised sets (**Rebuttal Figure 9B**).

Rebuttal Figure 9: Summary for 10,000 ancestry randomised runs of Model 1.

A) Odds ratios (OR) from Fisher exact test comparing the number of times we observed significant association with ancestry for the indicated ancestry pair compared to aggregate counts for the rest of ancestry pairs, across 10,000 permutations. B) Fold change estimates and 95% confidence intervals for permuted and the original data. In the permuted data, fold changes and standard errors were obtained by averaging each metric for their respective ancestry pair across 10,000 permutations. Asterisks indicate statistical significance at $p \leq 0.05$.

4.2 The statistical model employed by the authors uses a single parameter for the mean of the parental ages. This choice is hard to justify. It is plausible that age differences between parents are ancestry dependent. It is also known that maternal and paternal mutations have different distributions along the genome and different spectra. It is very possible that these effects are minor, but it would be better to see a model with maternal and paternal ages being separate covariates.

We agree that this warrants closer attention. We performed a more careful check on parental ages at conception for different ancestries and added the paternal and maternal age at conceptions as separate covariates in **Model 1** (which tests for ancestry differences in DNM counts) and **Model 2** (which tests for ancestry differences in spectra). These modified models gave similar results to those we presented originally (**Rebuttal Figure 1 and 3**), with the exception of the EUR and SAS pair, where differences in DNM counts are no longer significant. From this point, any references made to **Model 1** and **Model 2**, include the aforementioned modification. Also note that we have changed the statistical modelling approach for Model 2 to better account for the proportional nature of pyridine substitutions. Please refer to **response 2.4** for details in this analysis.

4.3 It would be of great interest to check if the spatial distributions of mutation along the genome (for example, clustered maternal mutations) show any differences by ancestry. This may shed some light on the underlying biology. I certainly appreciate that power can be an issue here.

We appreciate the reviewer's suggestion. This is an interesting question we were missing to address. We have checked this in our dataset and included this as part of **Supplementary Note #7**, and **Supplementary Figures 15-18**. Following a strategy used in published work ⁽⁵⁾, we divided the genome into 2Mb windows and then calculated the number of DNMs in each bin. We corroborated that we were able to replicate previous findings ⁵ regarding an increased number of DNMs occurring in the peri-telomeric regions of chr8 and chr16 (**Rebuttal Figure 10A-C**), which are mostly attributable to maternal DNMs (**Rebuttal Figure 10C**). We then reproduced this analysis while stratifying by ancestry. As raw counts per bin would reflect the sample size imbalance of this cohort (dominated by the EUR group) (**Rebuttal Figure 11**), we normalised DNM counts per bin per ancestry by the total DNMs detected in a given ancestry (and phasing group) (**Rebuttal Figure 12**). This visualisation reproduces the observations on maternal DNM excess in the chromosome 8 and 16 peri-telomeric regions, which seem consistent across all ancestry groups (**Rebuttal Figure 12C**). Ancestry-specific spatial clustering signals are difficult to ascertain due to the limited sample size for most non-EUR groups. In particular, it is difficult to assess the significance of individual peaks (e.g. total DNMs in chr22, **Rebuttal Figure 12A**), but the fact that none are apparent in the best-sampled ancestry group (EUR) suggests that more data would be necessary to confidently investigate these signals. We also note that very low DNM counts in windows next to centromeres and telomeres very likely reflect DNM callability in these regions.

A # of Total DNMs
(Window size 2Mb)

B # of Paternal DNMs
(Window size 2Mb)

Rebuttal Figure 10. Spatial distribution of GEL DNMs across the human genome (2 Mb binned).

Each point represents the sum of DNMs counts falling into a 2Mb window along a given chromosome. Representing counts A) per trio (i.e. unphased), B) paternally phased, and c) maternally phased. Red line represents the genome-wide DNM count per DNM category (i.e. per trio or parentally phased), while the dotted yellow line represents standard error. Excess DNM counts in peri-telomeric regions of chromosomes 8 and 16 are identified with green squares.

Rebuttal Figure 11. DNM counts (unphased) per ancestry group.

The top of each bar represents the number of DNMs (unphased) per each genetically inferred ancestry category

A # of Total DNMs
(Window size 2 Mb)

B # of Paternal DNMs
(Window size 2 Mb)

Rebuttal Figure 12. Spatial distribution of ancestry stratified GEL DNMs across the human genome (2 Mb binned).

Each point represents the fraction of phased DNMs counts for that ancestry group falling into a 2Mb window along a given chromosome. Representing counts A) per trio (i.e. unphased), B) paternally phased, and C) maternally phased. Dotted yellow line represents the genome-wide DNM count per DNM category (i.e. per trio or parentally phased). Excess DNM counts in peri-telomeric regions of chromosomes 8 and 16 are identified with black squares.

4.4. It is of interest whether the apparent ancestry differences may be due to early somatic mutations in parents rather than mutations in the germline. This is a subset of all mutations called *de novo*. Is it possible to see if the deviating classes of mutation calls have any notable shifts in coverage in parents?

We appreciate the reviewer's question. We are preparing a separate publication focusing on parental postzygotic mutations (PZMs) in the genomics England (GEL) family trios. This effort gave us several reasons to believe PZM may not be a particular problem for the analyses we presented. Most importantly, the *de novo* filtering pipeline applied to the DNMs we used for our analyses excluded DNM candidates with an ALT read coverage higher than 1 in either of the parents², so excluded many post-zygotic mutations. In our other work, we found that detecting PZMs in this cohort was particularly challenging due to the shallow sequencing depth of the cohort trios. Even when applying a specialised filtering pipeline specifically designed to ascertain parental PZM events out of all possible Mendelian inconsistencies (i.e. all DNM candidates prior any filtering), we were able to ascertain only ~1,900 high confidence parental PZM events cohort-wide. In spite of these priors, in response to the reviewer's comment, we tested whether putative PZMs were over-represented in a given ancestry category.

Around 11% of the DNMs we used in our analyses have an ALT coverage of 1 in one of the parents. We assumed that any of these could be potential PZM, and we tested whether these events were more common in any of the ancestry groups we studied. For this, we first classified all DNMs analysed in a given trio as putative PZMs (i.e. ALT read coverage == 1 in either of the parents) or true DNMs (i.e. ALT read coverage == 0 in both parents) and generated counts for these per ancestry. Then, for each ancestry, we asked whether PZM counts were significantly different in that ancestry as compared to the sum of the counts for each event in the rest of the ancestries (i.e. the aggregated counts across all other ancestries). We tested this in a Fisher exact test and found no significant evidence of enrichment (or depletion) of putative PZM events in any of the ancestry groups (Rebuttal Figure 13).

We have not added this result to the manuscript since we intend to address it more thoroughly in a separate publication focused on PZMs.

Rebuttal Figure 13: Fisher exact test odds ratios and 95% confidence intervals on counts of putative PZM vs true DNM events across ancestries.

4.5. I find the discussion of Harris and Pritchard, 2017 results somewhat misleading. I believe that there is a consensus in the community that the subtle SFS spike is due to a transient change in rate of specific mutations that was caused by a factor (more likely environmental and less likely genetic) that is not active at present. It has nothing to do with the present-day differences presented in this manuscript.

We thank the reviewer for their feedback. We believe it is valuable to include this comparison as we cannot rule out that the factors (either genetic or environmental) reported by Harris and Pritchard are still operating in modern populations. We have reworded our discussion to be clearer on the pitfalls of comparing our analyses against those of Harris and Pritchard:

Lines 246-255: “Interestingly, despite significant differences in overall DNM rates between AFR and non-AFR populations, no significant differences were found in their mutational spectra. This also contrasts with previous studies based on polymorphism data, where many of such differences were found 14. This discrepancy may be due to the fact that polymorphism data reflect mutations accumulated over thousands of generations, whereas our data only capture mutations from a single generation. Historical germline mutation changes may be influenced by environmental or genetic factors that have since changed, resulting in contemporary populations showing no such spectral differences. Furthermore, our ability to detect differences in DNM rate and spectra across ancestry pairs was likely impacted by the sample size differences between groups in our cohort(...).”

References

1. Zhou, H., He, K., Chen, J. & Zhang, X. LinDA: linear models for differential abundance analysis of microbiome compositional data. *Genome Biol.* **23**, 95 (2022).
2. Kaplanis, J. *et al.* Genetic and chemotherapeutic influences on germline hypermutation. *Nature* **605**, 503–508 (2022).
3. Caulfield, M. *et al.* National Genomic Research Library. Preprint at (2020).
4. Harris, K. & Pritchard, J. K. Rapid evolution of the human mutation spectrum. *Elife* **6**, (2017).
5. Jónsson, H. *et al.* Parental influence on human germline de novo mutations in 1,548 trios from Iceland. *Nature* **549**, 519–522 (2017).
6. Wang, R. J., Al-Saffar, S. I., Rogers, J. & Hahn, M. W. Human generation times across the past 250,000 years. *Sci. Adv.* **9**, eabm7047 (2023).
7. The 100,000 Genomes Project Pilot Investigators. 100,000 Genomes Pilot on Rare-Disease Diagnosis in Health Care — Preliminary Report. *N. Engl. J. Med.* **385**, 1868–1880 (2021).

The impact of ancestral, environmental and genetic influences on germline de novo mutation rates and spectra

We thank the reviewers again and are very pleased that they are satisfied with the revisions we have made. Reviewer 4 raises one additional concern, which we address below.

Reviewer #4 (Remarks to the Author):

The revised manuscript is an improvement. I am satisfied with the authors' response to all original comments. As a result of re-reading the manuscript, I have a new quick comment (see below). I apologize that it was omitted from the original review (unless I am missing this).

1) The manuscript states that the rate of de novo mutations in parent-child trios of African ancestry is slightly elevated compared to trios of other continental ancestry groups. The main source of false-positive mutation calls is undercalling very rare heterozygous alleles in parents. African ancestry individuals have more rare heterozygous alleles per genome. It would be great to see evidence that this effect is not driving the signal.

We emphasize that the DNM variant calling was done based on a multi-sample vcf that was constructed by family, rather than using all individuals in the dataset, as described here https://re-docs.genomicsengland.co.uk/de_novo_data/. Thus, the frequency of the allele at the population level will not affect the likelihood of a variant being called within the family. Hence, the fact that African ancestry individuals have more rare heterozygous alleles per genome should not affect the variant calling. We have added a sentence into the Results to explain this (lines 87:89):

"Furthermore, since the DNM calling was based on variant calls made only at the family level, the frequency of the variant across the cohort should not have affected sensitivity, so the preponderance of rare variants in individuals with African ancestry could not be driving our observation that they have more DNMs on average."

We have also added this detail into the Methods (lines 348-350):

"De novo variant calls (DNMs) per trio were generated using the Platypus variant caller⁴¹ based on multi-sample VCFs that were generated at the family level (https://re-docs.genomicsengland.co.uk/de_novo_data/)."

However, the reviewer is correct that the main source of false-positive mutation calls is under-calling heterozygous alleles in the parents. We addressed this in Supplementary Note 2, specifically in the following section, which we have edited to further clarify this issue (new additions in purple), and which we hope will satisfy the reviewer:

"Another indicator of potential mapping bias may be the parental coverage for the alternate site classed as "de novo" in the offspring. If a given ancestry group has higher mapping errors on average, or is associated with other issues affecting variant calling, variants which are present in the parents and passed on to the child may have low variant allele fraction (VAF), such that the heterozygous parental genotype is not called and these sites are erroneously called as DNMs. To check this, we calculated the mean parental variant allele fraction (VAF) ([maternal

*VAF + paternal VAF] / 2) at putative DNM sites, calculated the mean across DNMs per offspring, and then included this metric as a covariate in **Model 1** (main **Methods**). We found that this does not affect the original ancestry effect estimates (**Supplementary Figure 4**).*”

The impact of ancestral, environmental and genetic influences on germline de novo mutation rates and spectra

We thank the reviewer for their input and appreciate their concern regarding a possible source of artifacts, which we address below.

Reviewer #4 (Remarks to the Author):

"I appreciate the quick response to my remaining comment. I apologize again for this comment surfacing so late in the process (especially having been on the receiving end of such late comments). Saying that, I do not find the response convincing. There is no indication that calling per family rather than per batch solves the problem. First, individuals of African ancestry have more heterozygous sites. If there is a small fraction of false-positive de novo mutation calls due to undercalled hets in parents, it would masquerade as the difference in rate of de novo mutations between ancestry groups. Second, in many variant calling methods, allele frequency information indirectly leaks into family-based calls. We have never tested Platypus, but the error profile of trio-based calls using other methods (e.g. GATK) suggests that they remain sensitive to population allele frequency. The observed effect is subtle and is the main result of the paper. It would be helpful to make sure that it does not result of a variant calling artifact."

We agree that the larger number of heterozygous genotypes in individuals of African ancestries could, in theory, mean that a given rate of undercalled heterozygotes in the parents might appear as an increased number of (false) DNMs in the child. However we think this phenomenon, even if occurring, is unlikely to be affecting our results, for several reasons.

If there were false positive DNMs being called as a result of missed heterozygous calls in the parents, it is likely that, at these sites, the (truly) heterozygous parent would be called as homozygous reference but with a small number of reads carrying the alternate allele. This would manifest in those sites having a non-zero variant allele fraction (VAF) in the relevant parent (i.e. variant allele fraction = fraction of reads carrying the alternate allele). We explored this possibility in our previous response, referring to a section of **Supplementary Note 1**. We have now made a minor modification to this section of **Supplementary Note 1 (lines 960-975)**, to make clear that controlling for parental VAF is effectively controlling for the possibility of missed heterozygous sites in the parents. This section now reads:

*"Another indicator of potential mapping bias may be the parental coverage for the alternate site classed as "de novo" in the offspring. If a given ancestry group has higher mapping errors on average, **or is associated with other issues affecting variant calling**, variants which are present in the parents and passed on to the child may have low **but non-zero** variant allele fraction (i.e. **fraction of reads carrying the alternate (ALT) allele**, VAF), such that the heterozygous parental genotype is not called and these sites are erroneously called as DNMs. To check this, we calculated the mean parental variant allele fraction (VAF) ($[maternal\ VAF + paternal\ VAF] / 2$) at putative DNM sites, calculated the mean across DNMs per offspring, and then included this metric as a covariate in **Model 1 (main Methods)**. We found that this does not*

affect the original ancestry effect estimates (Supplementary Figure 3). This check also implies that our results are unlikely to be driven by the higher rates of heterozygous genotypes in individuals of African ancestries; if heterozygous genotypes in the parents are falsely called as homozygous reference, they are still likely to have low but non-zero VAF, and we have shown that controlling for average parental VAF does not affect our ancestry inferences. We describe additional investigation of this potential artefact below.”

Supplementary Figure 3 is reproduced at the bottom of this response document for convenience.

We have also carried out some new analyses to provide additional reassurance that ancestry-differential under-calling of heterozygous sites is unlikely to be affecting our results.

1. If this were the case, we would expect to see that in African-ancestry trios, a higher fraction of the DNMs passing filtering would have evidence of the ALT read in parents, corresponding to an uncalled heterozygous genotype. Our filtering pipeline already removed candidate DNMs with more than 1 ALT read in parents, but 11% of the DNMs we used in our analyses have an ALT coverage of exactly 1 in one of the parents. In our original response to reviewers (point 4.4 regarding putative post-zygotic mutations), we reported that these DNMs with 1 parental ALT read were not enriched in probands with any given ancestry group (Fisher's exact test $p > 0.05$). We have now additionally tested whether the parental variant allele fraction (VAF) distribution at such DNMs differs significantly between Africans and all other probands, using a Kolmogorov-Smirnov test, and found that it did not (Figure R1A below; $p = 0.65$). We also performed pairwise comparisons of parental VAF across all ancestries. We found subtle but significant differences only between the mean parental VAF of SAS/EUR and SAS/AMR ancestry pairs (**Figure R2B**, ANOVA/Tukey HSD post hoc test adjusted p value ≤ 0.05), which were not pairs for which we found significant DNM count differences. Repeating these analyses with all DNMs that passed our filtering (as opposed to those with 1 parental ALT read) found no significant differences in mean parental VAF between any ancestry groups. These results are consistent with the fact that, as we showed in our original submission and have noted again above, controlling for average parental VAF at the DNMs did not alter our conclusions about the ancestry differences. Given the relevance and value of these extra sanity checks for our main analyses, we have added these in our **Supplementary Note 1 (lines 976-999 for putative PZM differences; lines 1010-1025 for putative VAF differences)**, and have also referenced this in our main text (**lines 91-93**).

Figure R1. Distribution of parental VAFs for all DNMs passing stringent filters but with parental ALT count == 1 (n= 73,103). A) Distribution of parental VAFs for DNMs in AFR trios (is.afr == 1; n = 1,612) versus those in any other ancestry (is.afr == 0; n = 71,488). **B)** Pairwise comparison of parental VAFs stratified by ancestry. Top bars and asterisks indicate significant differences for the mapped ancestry pair (ANOVA/TukeyHSD test adjusted $p \leq 0.05$).

2. If under-calling of parental heterozygous sites were leading to inflated DNM counts in populations with higher heterozygosity, we would expect to see an association between the number of heterozygous sites in parents and the number of DNMs in their offspring within a given ancestry group, since the number of uncalled heterozygous sites is likely to be proportional to the number that have been called and passed QC.

To test this, we calculated the number of heterozygous genotypes (HETs) per parent for all trios used in our original DNM ancestry regression (n = 9,820). We calculated HET counts from variants called at the individual level that passed basic sample-level quality filters described elsewhere (https://re-docs.genomicsengland.co.uk/sample_qc/). We first checked that the heterozygosity per ancestry was similar to that reported for ¹ by calculating the HET/HOM ratio, a common measure of heterozygosity that is known to be ancestry specific ², and found that it was (Table R1).

Ancestry	GEL median HET/HOM	GEL SD	1000G median HET/HOM	1000G SD
AFR	1.95	0.112	2.00	0.04
AMR	1.6	0.116	1.63	0.11
EAS	1.33	0.042	1.31	0.04
EUR	1.56	0.024	1.55	0.02
SAS	1.58	0.105	1.57	0.03

Table R1: Heterozygosity (HET/HOM ratio) stratified by ancestry for GEL parents compared to 1000 Genomes (1000G) individuals ¹

We then tested the association between the number of DNMs per trio and the mean number of HETs per parent while stratifying by ancestry. To do this, we fitted a generalised linear regression model of the quasipoisson family in each ancestry group separately, while controlling for technical and biological covariates associated with the number of DNMs per trio as shown in **Model R2**:

$$\begin{aligned}
\text{trio DNM counts} = & \beta_0 + \text{mean parental nHETs} \cdot \beta_1 + \\
& \text{maternal age at conception} \cdot \beta_2 + \text{paternal age at conception} \cdot \beta_3 + \\
& \text{mean sequencing depth}_{\text{mother}} \cdot \beta_4 + \text{mean sequencing depth}_{\text{father}} \cdot \beta_5 + \\
& \text{mean sequencing depth}_{\text{offspring}} \cdot \beta_6 + \\
& \text{percent aligned reads}_{\text{mother}} \cdot \beta_7 + \text{percent aligned reads}_{\text{father}} \cdot \beta_8 + \\
& \text{percent aligned reads}_{\text{offspring}} \cdot \beta_9 + \\
& \text{median Bayes Factor per trio} \cdot \beta_{10} + \text{median VAF per trio} \cdot \beta_{11} + \varepsilon
\end{aligned}$$

where the *mean parental nHETs* represent the average number of HETs across both parents in a given trio. The rest of the covariates are the same as those outlined in **Model 1** in the main text of our manuscript.

We found that the mean parental nHETs were not associated with DNM counts in any of the ancestry groups (**Table R2**; p value > 0.05). Although the heterozygosity between parents of the same ancestry is expected to be similar, we also tested whether the number of HET sites in the maternal and paternal genomes contributed independently to the DNM counts per trio by including these metrics in **Model R2** instead of the mean parental nHET. We found no significant evidence of association between DNMs and either nHET metric (**Table R2**; p value > 0.05).

Ancestry	mean parental nHETs p value	maternal nHETs p value	paternal nHETs p value
AFR	0.15	0.10	0.40
AMR	0.57	0.88	0.72
EAS	0.34	0.37	0.86
EUR	0.61	0.99	0.50
SAS	0.89	0.51	0.42

Table R2: P-values for the association between parental DNMs per trio and number of heterozygous genotypes in the parents (nHETs) from a generalized linear model fitted to each ancestry group separately.

Again this finding suggests that the higher number of heterozygous genotypes in African-ancestry individuals is unlikely to be driving our finding of an excess of DNMs in this group.

We have added these two new analyses to **Supplementary Note 1 (lines 1026-1056)**, in the section on “*Biases due to ancestrally-differential rates of missed constitutive heterozygous calls in parents*”, and signposted these additional analyses **at line 91** in the main text. We note that we are effectively testing the same thing (i.e. whether African-ancestry parents tend to have higher VAF at DNM sites) in different ways in different parts of **Supplementary Note 1**, in the context of checking for (a) ancestry-related mapping bias, (b) ancestral differences in rates of post-zygotic mutations, and (c) ancestral differences in heterozygosity. We have chosen this structure to signpost our responses to the specific reviewer queries about (b) and (c).

“Depending on the stringency of the calling procedure, it may generate either more false-positive calls in families of African ancestry or more false-negative calls in families of other ancestries. It would be very helpful to obtain an upper bound estimate of the error rate by ancestry. If the authors have access to multi-generational families, it may provide the best way to analyze the errors due to undercalled hets. Otherwise, testing the pipeline on gold standard datasets and plotting error rates by coverage and allele frequency would be helpful.”

Unfortunately there are no multi-generational pedigrees with appropriate samples available in the 100,000 Genomes project for us to use to test the pipeline. We are not sure which “gold-standard datasets” the reviewer is recommending. The most commonly-used gold-standard is Genome in a Bottle but this only includes three trios, one of European ancestry (NA12878 from the CEPH pedigrees), one of Ashkenazi Jewish ancestry and one of East Asian ancestry. Hence, this would not be suitable for addressing this concern about under-calling of parental heterozygotes in Africans. Additionally, the NA12878 trio was already used for benchmarking the Platypus DNM pipeline in the original publication³. In this, Platypus was found to call 49/49 germline DNMs in the NA12878 trio which were considered true positives according to Conrad et al. (Nature Genetics, 2011), of which 46 passed its Bayesian filter.

Rimmer et al. also attempted to experimentally validate DNM calls in another trio, and managed to validate 63 out of the 68 attempted. Of the five that did not validate, “one was due to a missed parental variant due to read filtering on the basis of mapping quality” (hence, truly heterozygous in the parents), and the others were likely due to mapping artifacts (reads of low MAPQ lying in repetitive elements). Thus, we cannot rule out that a small fraction of the DNMs that have passed our filtering are false positives due to missed ALT reads from the parents. However, we believe the analyses presented above show that this is unlikely to be driving the ancestry differences we see.

We considered scrutinizing a subset of DNMs in IGV to look for artefacts (such as reads with the ALT allele in parents which failed the MAPQ threshold and were thus not ‘seen’ by the variant caller) which might systematically differ in frequency between ancestry groups. However, since the average baseline DNM count differed by a maximum of ~4% between ancestry groups (specifically between AFR and EUR), if the false discovery rate in these DNM calls is similar to rate of ~7% estimated in the Platypus paper (5/68 DNMs failed validation experiments), then even if all of this ancestral difference is driven by false positives, we would need to look at 3000-4000 DNMs in each ancestry group to establish with statistical significance whether an excess of errors in the AFR group could explain it. We believe this is not a reasonable task and that the analyses presented above are sufficient demonstration that artefacts are unlikely to be driving our signal.

References

1. Samuels, D. C. *et al.* Heterozygosity ratio, a robust global genomic measure of autozygosity and its association with height and disease risk. *Genetics* **204**, 893–904 (2016).
2. Wang, J., Raskin, L., Samuels, D. C., Shyr, Y. & Guo, Y. Genome measures used for quality control are dependent on gene function and ancestry. *Bioinformatics* **31**, 318–323 (2015).
3. Rimmer, A. *et al.* Integrating mapping-, assembly- and haplotype-based approaches for calling variants in clinical sequencing applications. *Nat. Genet.* **46**, 912–918 (2014).

“Supplementary Figure 3. Ancestry associations with DNM counts before and after controlling for covariates capturing potential artifacts.

Bar colours represent a regression model for DNM rate \sim ancestry, including all of the original covariates in Model 1 plus an extra potential artifactual source. The extra covariates included were average average mismatches per read per trio (avg avg NM), maximum average mismatches per read per trio (max avg NM), mean parental variant allele fraction per trio (parental VAF), or number of potentially deleterious de novo variants (n deleterious DNMs). The grey bar corresponds to the original Model 1 outlined in Methods. Asterisks indicate ancestry effect significance at 5% FDR ($p_{\text{adjusted}} \leq 0.05$).”

The impact of ancestral, environmental and genetic influences on germline *de novo* mutation rates and spectra

We thank the reviewer for their input and we are pleased to see they are happy with our latest response to their comments.

Reviewer #4 (Remarks to the Author):

The analysis of VAF and inclusion of heterozygosity in the model do suggest that the effect is either absent or minor. I do not have other concerns. The authors may choose to show that the difference between populations persist even after the exclusion of all ALT==1 sites in parents.

We appreciate the reviewer's suggestion and we have included this as part of our supplementary note 1 section c, and have illustrated this analysis in the panel C from our supplementary Figure 5. We show that our reported ancestry associations to DNM rates remain at similar effect sizes as our original models. However, excluding DNMs per trio resulted in slightly wider confidence intervals that caused 2 of our associations to only reach nominal significance. We believe this may be due to a reduction in power.

To respond to the authors' question about multigenerational data, there is a multigenerational public dataset of a large Utah family. It does not allow for the comparisons between ancestry groups but does allow for estimation of the number of over/under-called *de novo* mutations.

We thank the reviewer for the suggestion, we will consider the inclusion of this dataset for benchmarking GEL DNM counts in future work.